# Sonogenetic control of multiplexed genome regulation and base editing

Pei Liu [1,2], Josquin Foiret [1], Yinglin Situ[2], Nisi Zhang [1], Aris J. Kare [1,2], Bo Wu[1], Marina N. Raie [1], Katherine W. Ferrara [1] ✉ & Lei S. Qi [2,3,4] ✉

Manipulating gene expression in the host genome with high precision is crucial for controlling cellular function and behavior. Here, we present a precise, non-invasive, and tunable strategy for controlling the expression of multiple endogenous genes both in vitro and in vivo, utilizing ultrasound as the stimulus. By engineering a hyper-efficient dCas12a and effector under a heat shock promoter, we demonstrate a system that can be inducibly activated through thermal energy produced by ultrasound absorption. This system allows versatile thermal induction of gene activation or base editing across cell types, including primary T cells, and enables multiplexed gene activation using a single guide RNA array. In mouse models, localized temperature elevation guided by high-intensity focused ultrasound effectively triggers reporter gene expression in implanted cells. Our work underscores the potential of ultrasound as a clinically viable approach to enhance cell and gene-based therapies via precision genome and epigenome engineering.

Cellular engineering and reprogramming hold great promise in advancing therapeutic applications and tackling various diseases. Regulation of genes of interest in vivo at a precise location and time improves treatment efficiency and safety[1–5]. Several methods have been developed for the efficient control of gene expression in vitro but most show limited applications in vivo. For example, drug-inducible gene expression is commonly used for cultured cells or mouse models but problems such as toxicity and lack of cellular, spatial, and temporal specificity remain[6,7]. Optogenetics using light is attractive due to its high spatiotemporal resolution; however, its poor penetration and high invasiveness in vivo hinders its application beyond preclinical models[8,9]. Other stimuli such as radiofrequency radiation and magnetic field not only lack specificity but also require the application of nanomaterials to interact with cells of interest[10,11]. Thus, there is a high demand for the development of novel physical tools, enabling more precise, non-invasive, and tunable control of cells in vivo.

Ultrasound has been primarily employed as an imaging and diagnostic modality, while its therapeutic benefits have yet to be fully explored. Sonogenetics is an emerging field in which ultrasound is used to regulate cellular function and activity[12–14]. While most studies focus on the mechanical effect of ultrasound on calcium fluctuations and neuromodulation through ion channels such as TRP-4[15], MscL[16,17], and TRP-A1[18], few studies directly link the ultrasound stimulus with endogenous gene expression in the genome. Recently, it's shown that mechanosensitive Piezo1 can be stimulated by ultrasound, triggering chimeric antigen receptor (CAR) expression through a NFAT promoter; however, microbubble-induced microstreaming is required for channel activation and limits the application of this system in vivo[19]. Besides the mechanical effect, ultrasound with a higher frequency or intensity can generate heat. This has been used to activate a heat shock promoter to drive ectopic CAR expression in T cells to enhance anti-tumor activity[20]. Alternatively, a thermo-sensitive bacterial repressor and genetic circuit induced ultrasound-dependent GFP reporter activation in mice[21,22]. As genome engineering is becoming a pillar for gene- and cell-based therapies, stimuli-dependent precise control of multiple endogenous genes is needed to ensure tissue selectivity and specificity. On the other hand, previous strategies require cloning each new gene of interest under

[1]Molecular Imaging Program at Stanford (MIPS), Department of Radiology, School of Medicine, Stanford University, Stanford, CA, USA. [2]Department of Bioengineering, Stanford University, Stanford, CA, USA. [3]Sarafan ChEM-H, Stanford University, Stanford, CA, USA. [4]Chan Zuckerberg Biohub – San Francisco, San Francisco, CA, USA. ✉e-mail: kwferrar@stanford.edu; slqi@stanford.edu

the promoter and therefore multiplexed gene expression control has been challenging. As a result, there is a need to develop new approaches to employ ultrasound as a non-invasive and precise physical stimulus to provide a more versatile and efficient platform for multiplexed endogenous gene modulation.

In this work, we present a strategy that relies on the thermal effect of focused ultrasound to control endogenous gene expression in vitro and in vivo by an improved CRISPR-Cas system. Expression of the Cas protein and its effector can be modulated under a heat shock promoter (HSP), which is rapidly responsive to ultrasound-induced heating. One group has recently engineered HSP-mediated thermo-sensitive dCas9 with effectors and demonstrated in vivo transcriptional repression of individual genes by KRAB-dCas9 using near-infrared (NIR) excitation and implanted gold nanorod[23]. Compared to this work, our strategy uses the Cas12a family of proteins for multiplexed gene activation or base editing via ultrasound-mediated control, which is more penetrative and clinically relevant compared to NIR. Furthermore, our approach is more scalable, as any one or more genes of interest can be flexibly targeted by a compact guide RNA array, which can be conveniently synthesized for easy delivery compared to the traditional promoter-based cloning (i.e., each gene requires a different promoter). We demonstrate the use of a transcriptional activator or base editor to achieve diverse types of genome engineering. Our approach for thermally mediated CRISPR-Cas activation shows effective gene expression enhancement in various cell types including primary cells. Finally, we establish the feasibility of focused ultrasound to modulate multiple endogenous genes (four genes at a time) in vitro, as well as ultrasound-mediated gene activation in vivo.

## Results

### HSP-hyperdCas12a-miniVPR activates GFP reporter cells in vitro

To achieve gene expression control with CRISPR-Cas, Cas12a was selected due to its smaller size (3.6 kb) and the ability to process a compact guide RNA (gRNA) array for multiplexed control of genes[24,25] (Fig. 1a). A recently engineered hyper-efficient Cas12a, termed hyper-Cas12a, a variant of *Lachnospiraceae bacterium* Cas12a with higher efficiency for both in vitro and in vivo applications, was selected[26]. For ultrasound-mediated thermal activation, the human HSP6A (HSP70) promoter was used since it has been well-characterized to respond to focused ultrasound[27,28]. We first fused the nuclease-dead hyperdCas12a to a miniaturized VPR (miniVPR) for transcriptional activation and cloned it under a truncated HSPA6 promoter[29] (Supplementary Fig. 1a). As previously described, the truncated HSPA6 promoter is half the size of the full-length promoter but exhibits much improved thermal response. The functionality was tested in a doxycycline-inducible HEK293T TRE3G-GFP reporter line with a gRNA (crTet) targeting the TRE3G promoter such that GFP expression is activated upon hyperdCas12a-miniVPR expression (Fig. 1b).

When cells were exposed to elevated temperatures from 40 °C to 43 °C for 30 min in a thermal cycler, we observed increased mean GFP fluorescence intensity and percentage of GFP-positive cells (Fig. 1b, Supplementary Fig. 1b). The greatest fold change was achieved at 43 °C for 30 min, with over 60-fold activation compared to unheated cells (i.e., 37 °C). Nonetheless, cell viability decreased to approximately 65% at 43 °C while cells tolerated lower temperatures well (Supplementary Fig. 1b). Notably, even when the temperature was raised by 3 °C to 40 °C, we began to see GFP expression, suggesting that it was a highly sensitive system. With each 1 °C increment from 40 °C to 43 °C, we

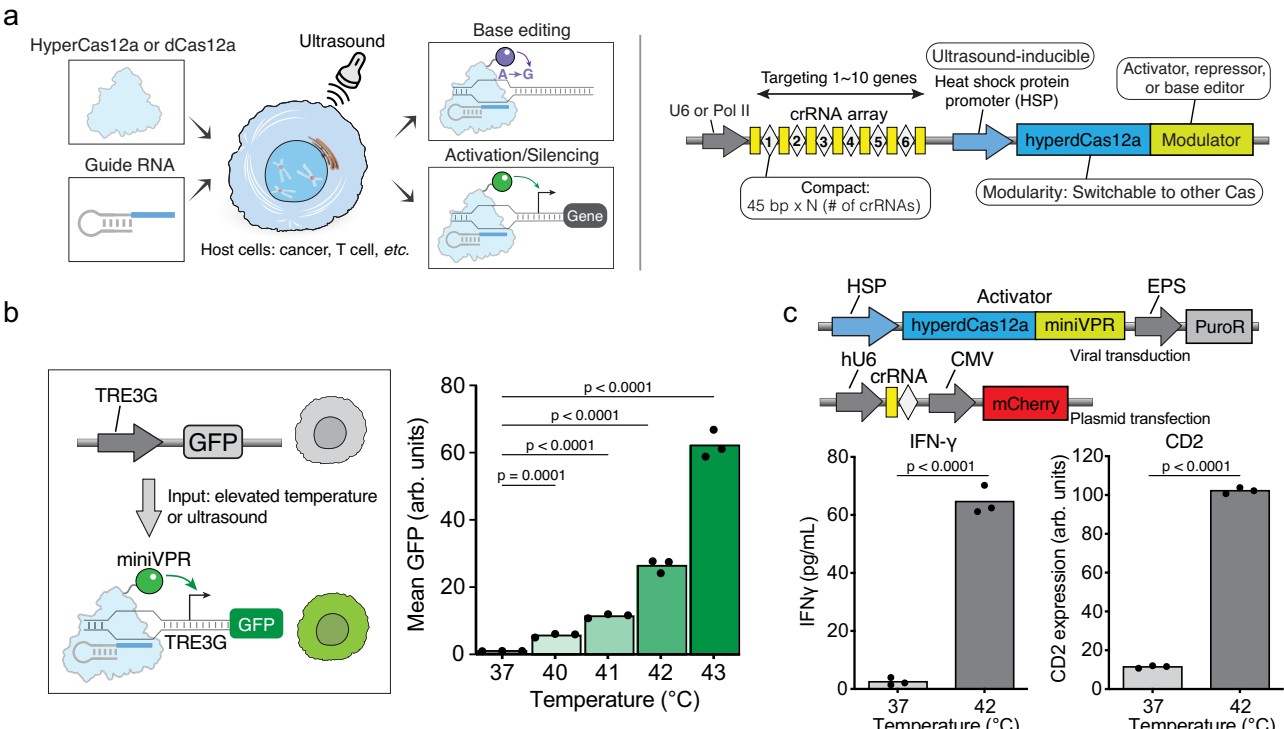

**Fig. 1 | Heat-induced hyperdCas12a for gene activation and base editing in HEK cells. a** Schematic showing ultrasound-induced hyperCas12a protein and effector expression for versatile genome engineering. **b** Left: schematic showing GFP reporter activation in HEK293T cells upon heat treatment. Right: normalized mean GFP fluorescence intensity measured by flow cytometry when transduced cells were treated at different temperatures for 30 min. All measured values were normalized to the average intensity of samples treated at 37 °C. %GFP+ cells are shown in Supplementary Fig. 1b. **c** Top: constructs used in testing endogenous gene activation. Puromycin was used to create a stable cell line expressing heat-inducible hyperdCas12a-miniVPR. Plasmids containing gRNAs were transfected with > 90% of cells expressing mCherry/guide. Bottom: heat-induced activation of endogenous IFN-γ measured by ELISA and activation of CD2 measured by immunostaining. All data are shown for 3 independent biological replicates. Source data in this figure are provided as a Source Data file. An unpaired two-sided t-test was used for statistical analysis and *p*-values are presented in Supplementary Table 3.

obtained a range of 5.7- to 62.2-fold increase in mean fluorescence intensity, demonstrating the potential of precisely monitoring gene expression by simply changing the thermal input. The extent of GFP activation was also affected by the duration of heating, as longer duration at 42 °C and 43 °C improved the activation effect (Supplementary Fig. 1c). Based on the Sapareto-Dewey thermal dose equation[30], we converted the thermal input, comprising temperature and duration, into cumulative equivalent minutes at 43 °C. Both mean GFP fluorescence intensity and the percentage of GFP-positive cells reached a plateau as the thermal input increased while cell viability decreased (Supplementary Fig. 1c, d).

We conducted kinetics experiments in HEK293T TRE3G-GFP reporter cells to monitor the time-dependent activation and cessation using the HSP-hyperdCas12a-miniVPR system. We heated the cells at 43 °C for 15 min and characterized GFP activation at different time points until the cells reached confluency. We observed rapid GFP activation in 8 hr after heat induction. Gene activation reached maximum after 24 to 30 hr and started to decline to close to baseline at 80 hr after the treatment (Supplementary Fig. 2a). To characterize basal HSP-hyperdCas12a-miniVPR expression without heat treatment, we cultured and maintained the reporter cells for 3 weeks. Over this period, the mean fluorescence intensity or percentage of GFP-positive cells did not increase, suggesting that there was no leaky hyperdCas12a-miniVPR expression from the HSP promoter (Supplementary Fig. 2b).

## HSP-hyperdCas12a-miniVPR activates a variety of endogenous genes upon heat treatment

To establish the feasibility of our approach for thermal-mediated endogenous gene activation, a series of gRNAs targeting the promoter regions of various genes were tested. We first generated a HEK293T cell line stably expressing HSP-hyperdCas12a-miniVPR with puromycin selection (Fig. 1c). We chose five endogenous genes with verified gRNAs, including interferon-gamma (*IFNG*), interleukin 2 (*IL2*), and C-

X-C motif chemokine ligand 10 (*CXCL10*), which are important cytokines in modulating immune cell activity, as well as C-X-C chemokine receptor type 4 (*CXCR4*) and cluster of differentiation 2 (*CD2*), which are cell surface proteins. After transfecting the plasmid encoding gRNA, the cells were heated at 42 °C for 30 min and protein expression was quantified 24 hr later by enzyme-linked immunosorbent assay (ELISA) for secreted cytokines or by immunostaining for cell surface proteins. Gene activation and protein expression increased for all selected genes following heat treatment (Fig. 1c, Supplementary Fig. 3). These results indicate that we can flexibly target endogenous genes of interest by selecting a specific guide RNA in a heat-responsive manner.

## HSP-hyperdCas12a-miniVPR activates multiple genes simultaneously with one stimulus

We next designed a gRNA array targeting multiple genes, harnessing the capability of Cas12a in gRNA array processing[24,25]. We reasoned that heat-induced expression of hyperdCas12a-miniVPR can efficiently process the crRNA array into individual guides, thus activating multiple genes simultaneously. Using a gRNA array containing one guide for *IFNG* and two guides for *IL2*, the cells were transfected, treated, and analyzed as abovementioned (Fig. 2a). Upon heat treatment, enhanced levels of both IFN-γ and IL2 were detected. Compared to samples transfected with single guides, cytokine production moderately decreased in the samples with the guide array (Figs. 1c, 2a, Supplementary Fig. 3), which is likely due to inefficient gRNA array processing. We next generated a gRNA array containing five tandem guides targeting four endogenous genes, including *IFNG, IL2, IL18 and IL7* (Fig. 2b). To ensure efficient gRNA processing, we used an optimized repeat sequence in the gRNA array[31]. After transfecting the gRNA array and conducting heat shock in a thermocycler, we extracted mRNA and quantified the expression of these genes by RT-qPCR. We observed a significant increase in the expression (tens to hundreds of fold of activation) for all four genes, compared to the non-treated samples,

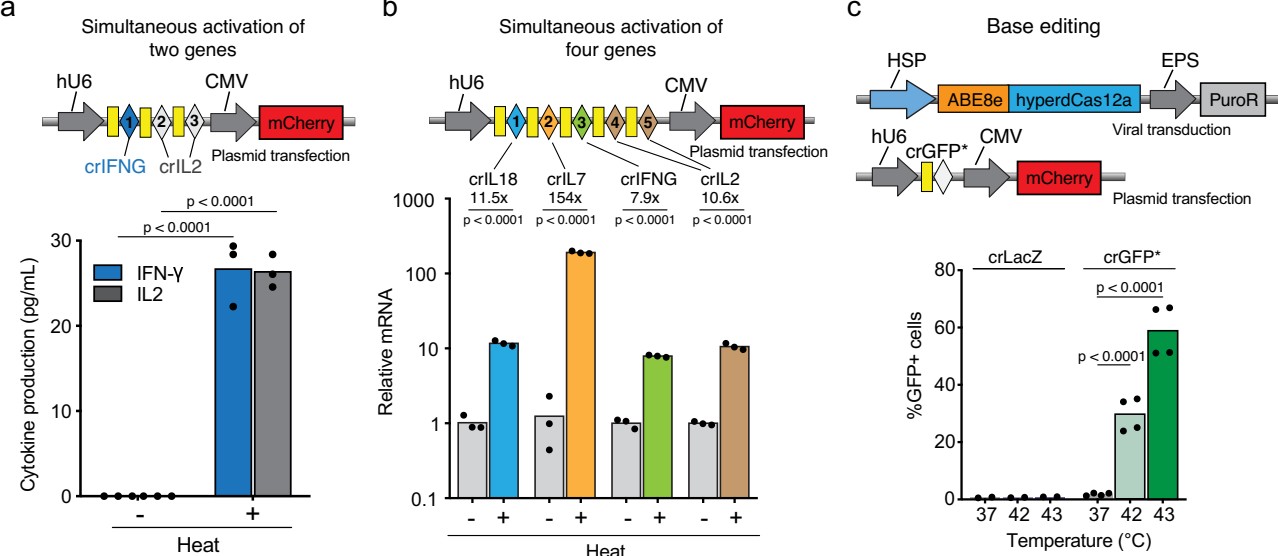

**Fig. 2 | Heat-induced hyperdCas12a for multiplexed gene activation and base editing in HEK cells. a**, Top: construct of gRNA array containing 3 guides targeting *IFNG* and *IL2*. Bottom: simultaneous activation and expression of both IFN-γ and IL2 measured by ELISA 24 hr after heat treatment at 42 °C for 30 min. **b** Top: construct of gRNA array containing 5 guides targeting *IL18, IL7, IFNG,* and *IL2*. Bottom: simultaneous activation of four genes measured by RT-qPCR after heat treatment at 43 °C for 15 min. **c** Top: constructs used in HEK293T cells with a mutant GFP* reporter containing a premature stop codon. Heat-inducible Cas was lentivirally transduced and selected by puromycin. Guide RNA was transfected. Bottom, the

percentage of GFP positive cells measured by flow cytometry 24 h after cells were treated at 42 or 43 °C for 30 min. %GFP+ cells were calculated as the percentage of GFP-positive cells in the mCherry-positive population (i.e. cells that express the guide). crLacZ: non-targeting guide. crGFP*: guide targeting the stop codon region in the GFP* reporter. Mean GFP values are shown in Supplementary Fig. 4b. All data are shown for 3 or 4 independent biological replicates. Source data in this figure are provided as a Source Data file. Unpaired two-sided t-test was used for statistical analysis and *p*-values are presented in Supplementary Table 3.

suggesting that the expression of multiple genes can be modulated simultaneously with one single stimulus (Fig. 2b). Together, these data confirm the scalability of our approach, as a single thermal-based strategy can be used to activate many genes of interest with high versatility and efficiency. This is particularly relevant to cellular engineering and reprogramming, where the modulation of multiple genes within a biological pathway is often essential to attain the desired cell function.

## HSP-ABE8e-hyperdCas12a inducibly and efficiently edits genes of interest upon heat treatment

The ability to edit the genome DNA sequence in selected cells is a major need in genome engineering. We hypothesized that HSP-hyperdCas12a would allow us to achieve inducible base editing in cells, thus offering other modes of genome manipulation (Fig. 1a). This idea was tested by fusing the nuclease-dead hyperdCas12a to a base editor ABE8e (Supplementary Fig. 4a). ABE8e is a recently evolved adenosine base editor with superior performance to convert A: T base pair to G: C and is compatible with hyperdCas12a[32]. To assess the efficiency of heat-inducible A to G modification, a HEK293T reporter cell line containing a premature stop codon in GFP (termed GFP*) was used (Supplementary Fig. 4a)[32]. Successful A to G editing corrects the stop codon and enables translation of the fluorescent protein. HSP-ABE8e-hyperdCas12a and a gRNA (crGFP*) targeting the region containing the premature stop codon were used (Fig. 2c). Compared to cells expressing a non-targeting guide (crLacZ) or cells without heat treatment, heat-induced ABE8e-hyperdCas12a expression efficiently corrected the premature stop codon and restored GFP fluorescence in almost 60% of the cells (Fig. 2c, Supplementary Fig. 4b). Notably, the performance of our HSP-inducible base editor is comparable to ABE8e-hyperdCas12a constitutively expressed under the SFFV promoter (Supplementary Fig. 4c). While our observations over a three-week period did not reveal leaky HSP-Cas expression or target gene activation in the absence of heat treatment, it is possible that long-term accumulation of low amplitude leakiness of the base editing system may lead to more pronounced, permanent changes on the genome. However, this can be mitigated by titrating the expression level of HSP-Cas in transduced cells to minimize the background expression at physiological temperatures.

## HSP-hyperdCas12a-miniVPR inducibly activates multiple endogenous genes in primary T cells under heat treatment

Engineered T cell therapies such as chimeric antigen receptor (CAR) T cells have seen remarkable progress in tackling many diseases including cancer[33,34]. Despite their ground-breaking successes against blood malignancies, challenges remain to treat solid tumors. A major hurdle for both CAR-T cells and native T cells is insufficient T cell activation due to a paucity and heterogeneity of tumor antigen expression, immunosuppressive tumor microenvironment, and rapid T cell exhaustion[33–35]. On the other hand, non-specific immune cell activation causes on-target, off-tumor adverse effects, leading to severe problems like neurotoxicity and cytokine release syndrome. Thus, there is a high demand for the development of more precise, non-invasive, and tunable control of engineered immune cells in vivo. Ultrasonic control of endogenous gene expression in T cells after cell infusion into the body offers new opportunities to manipulate cell activity with high spatial and temporal precision at the local sites of solid tumors[20]. We hypothesize that our strategy has the potential to enhance T cell activity through a heat-induced CRISPR-Cas system both in the context of CAR-T cells and in the context of wild-type T cell receptors.

We first evaluated the performance of HSP-hyperdCas12a for inducibly activating endogenous genes in Jurkat T cells. We transduced HSP-hyperdCas12a-miniVPR and gRNA targeting endogenous genes including *IFNG* or *IL2*. After thermal induction, we measured the

production of cytokines by ELISA and saw an approximate 10-fold increase in protein expression, compared to untreated cells (Fig. 3a). We further verified that the HSP-ABE8e-hyperdCas12a base editor could efficiently correct the premature stop codon by A to G editing in Jurkat cells (Fig. 3b, Supplementary Fig. 5a). In this experiment, almost 40% and 60% of cells expressed GFP after heat-induced base editing at 42 °C or 43 °C, respectively (Fig. 3b). These results highlighted the versatility of our approach and demonstrated that the HSP-hyperdCas12a system could be generalized to T cell lines.

We next tested whether our system achieved heat-induced reporter or endogenous gene activation in primary human T cells. We co-transduced pTRE3G-GFP, HSP-hyperdCas12a-miniVPR, and gRNA into primary T cells (Fig. 3c). After puromycin selection, we heated T cells at 42 °C or 43 °C for 15- or 30-min. Cell viability was reduced to 30% with treatment at 43 °C for 30 min (Supplementary Fig. 5c). When reducing the treatment duration to 15 min, we observed a 22-fold activation of fluorescence intensity compared to the untreated control, with 35% GFP positive cells and improved cell viability of 75% (Fig. 3c, Supplementary Fig. 5b, c). This condition of heating at 43 °C for 15 min is consistent with previous experimental protocols[20,36] and was therefore used for all T cell heat experiments.

To test the activation of endogenous genes, we chose to target the *FOXQ1* gene, which encodes a transcription factor that has been shown to promote T cell proliferation and activation[37]. When primary T cells transduced with gRNAs targeting the promoter of the *FOXQ1* gene were heated at 43 °C for 15 min, we observed more than 100-fold increase in mRNA expression of *FOXQ1*, as measured by RT-qPCR (Fig. 3d). These experiments together confirm that our HSP-hyperdCas12a system achieves efficient activation of reporter and endogenous genes in both T cell lines and primary T cells.

## High intensity focused ultrasound (HIFU) triggers efficient HSP-hyperdCas12a activities in vitro

Ultrasound imaging has been employed clinically as a non-invasive and effective diagnostic modality to image tissue with high spatial and temporal resolution. In addition, therapeutic ultrasound, using greater transmitted intensity or a lower center frequency has mechanical and thermal effects on isolated cells and tissue deep within the human body[38]. High-intensity focused ultrasound (HIFU) can raise the temperature within a defined region by tens of degrees within seconds; here, our goal was to raise the temperature to 43 °C and maintain this small temperature increase over time in a carefully controlled fashion. To test if focused ultrasound-induced mild hyperthermia can activate HSP-hyperdCas12a and subsequent gene expression, we first conducted a proof-of-concept experiment using fluorescent reporter cells. Using the abovementioned HEK293T TRE3G-GFP reporter cells transduced with HSP-hyperdCas12a-miniVPR and TRE-targeting gRNA (Supplementary Fig. 1a), we tested the effect of HIFU on GFP activation.

We first set up a dual-transducer system, which allowed us to visualize the sample while delivering focused ultrasound to the exact sample of interest (Fig. 4a, Supplementary Fig. 6a). The ultrasound setup employed for ultrasound-guided focused ultrasound (USgFUS) comprised a research ultrasound platform capable of both imaging and therapy and a dedicated small animal 1.5-MHz 128-element therapeutic array capable of three-dimensional beam steering[39]. An imaging array was located at the center of the therapeutic array for image guidance and treatment planning. The treatment sequence allowed precise targeting to minimize heat deposition outside the treated volume. Before the treatment, a needle thermocouple was inserted next to the tip of the tube containing cells to provide temperature feedback. The measured temperature was fed to a Proportional Integral Derivative (PID) controller that modulated the output of the ultrasound system to maintain the requested temperature throughout the experiment (Fig. 4a, Supplementary Fig. 6a, b). To reduce the acoustic pressure and maintain a homogeneous heating volume, the

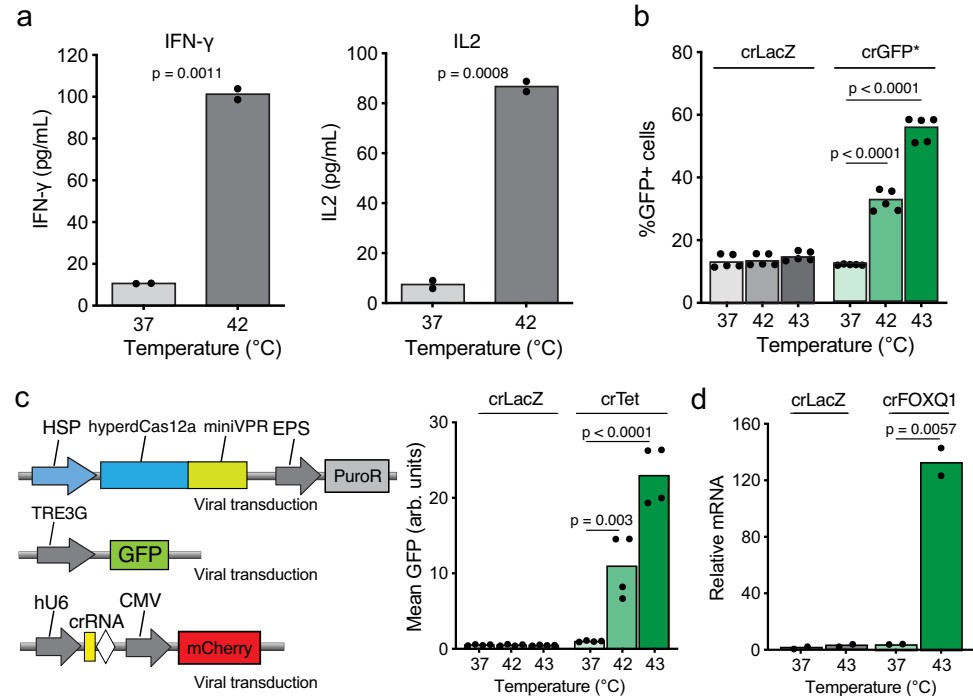

**Fig. 3 | Heat-induced hyperdCas12a for gene activation and base editing in Jurkat T cells and primary T cells. a** Heat-induced activation of endogenous IFN-γ and IL2 in Jurkat cells measured by ELISA. **b** Percentage of GFP positive (base-edited) Jurkat cells measured by flow cytometry 24 hr after cells were treated at 42 °C or 43 °C for 30 min. %GFP+ cells were calculated as the percentage of GFP-positive cells in the mCherry-positive population (i.e., cells that express the guide). crLacZ: non-targeting guide. crGFP*: guide targeting the stop codon region in the GFP* reporter. Mean GFP values are shown in Supplementary Fig. 5a. **c** Left: constructs used to transduce primary human T cells. Puromycin was added two days after transducing HSP-Cas construct and was incubated for two days to select for expressing cells. Right: mean GFP fluorescence intensity measured by flow cytometry when transduced T cells were treated at 42 °C for 30 min or at 43 °C for 15 min. crLacZ: non-targeting guide. crTet: guide targeting the TRE3G promoter. % GFP+ cells are shown in Supplementary Fig. 5b. **d** Endogenous gene activation in T cells transduced with a guide RNA targeting the FOXQ1 transcription factor. All data are shown for 2-5 independent replicates. Source data in this figure are provided as a Source Data file. An unpaired two-sided t-test was used for statistical analysis and *p*-values are presented in Supplementary Table 3.

HIFU array was phased to generate 8 simultaneous foci[40]. This method has the advantage of reducing the acoustic pressure in the tissue and providing a homogenous heat deposition in the sample.

From the thermocouple measurement, ultrasound with a high intensity increased the temperature from 35 °C to the designated 43 °C within 2 minutes and with minimal temperature fluctuation (within 0.2 °C) throughout the 15-min treatment duration (Supplementary Fig. 6b). Both GFP fluorescence intensity and percentage of GFP positive cells increased significantly compared to the non-treated control, though the activation effect was less than thermal activation in a thermocycler (Fig. 4b, Supplementary Fig. 6c).

We next tested the gRNA array targeting four endogenous genes in HIFU-treated cells. We observed substantial expression of *IFNG*, *IL2*, *IL18* and *IL7* in HIFU-treated cells, with 4.5-, 8.8-, 3.6-, 44.0- fold increase respectively, compared to the untreated cells (Fig. 4c). Using the ABE8e-hyperdCas12a base editor, ultrasound efficiently corrected the premature stop codon in the GFP reporter, resulting in 15% GFP positive cells (Fig. 4d, Supplementary Fig. 6d). A side-by-side comparison of HIFU and treatment in the thermal cycler indicated that HIFU-treated gene activation and base editing were less efficient (Fig. 2b, Fig. 4b–d, Supplementary Fig. 6c, d). Although the samples were raised to the same temperature with HIFU- or thermocycler-based treatment, there was a slight decrease in cell viability with HIFU treatment, likely due to the mechanical effects of acoustic waves on isolated cells, which possibly explained the weaker activation (Supplementary Fig. 6c). Collectively, these data demonstrate that HSP-hyperdCas12a is responsive to focused ultrasound and achieves ultrasound-mediated gene expression regulation and genome editing.

## HIFU triggers HSP-hyperdCas12a-miniVPR for gene activation in vivo

We next assessed whether HIFU activated genes in vivo. The engineered HEK293T cells were injected bilaterally into the flank of J:Nu immunocompromised mice (Fig. 5a). When the tumor size reached 4-5 mm, we conducted HIFU treatment on one tumor while the other tumor remained as the untreated control (Fig. 5a). The needle thermocouple was inserted subcutaneously between the tumor and the body wall. Once the thermocouple was in place, the animal was positioned on its side with the tumor facing the therapeutic array.

With our focused ultrasound setup, the tumor was visualized while the acoustic focus was localized to the center of the tumor for heating (Fig. 5c). We set the ultrasound system to increase the tumor temperature to 43 °C for a duration of 15 min. Careful planning of the acoustic energy output and targeting minimized off-target energy deposition, resulting in no apparent damage to the mouse skin or tissue. The tumors were extracted 24 hr after HIFU treatment and analyzed by flow cytometry for mCherry reporter activation (Fig. 5b). In the untreated tumor, we observed little mCherry expression while the treated tumor showed a significant increase in mCherry positive cells (Fig. 5d). The variation of reporter activation across the animals probably arises from varied efficiency of ultrasound absorption and heat dissipation in the tumor. Nonetheless, this in vivo HIFU experiment has verified that focused ultrasound is compatible with CRISPR/Cas tools in animal models. We hypothesize that by changing the guide RNA, we can non-invasively activate any gene or combination of genes of interest at specific locations in vivo with controlled timing through a guided HIFU treatment.

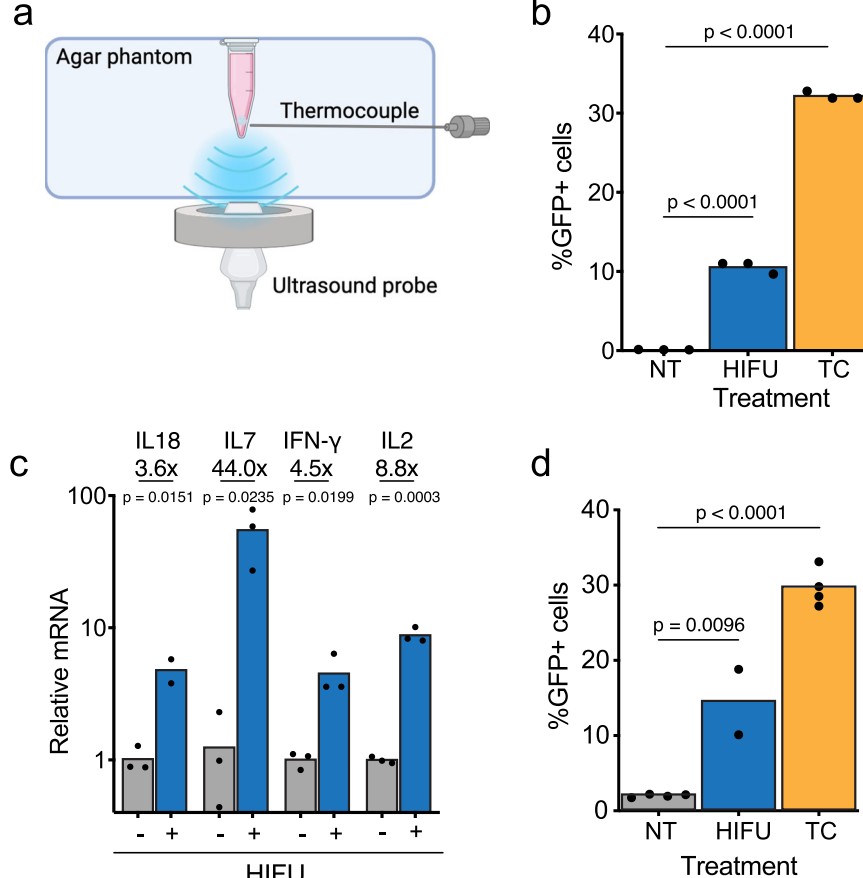

**Fig. 4 | High-intensity focused ultrasound triggers HSP-hyperdCas12a activities for multiplexed gene activation and base editing in vitro. a** Ultrasound-guided focused ultrasound (USgFUS) setup: schematic showing in vitro HIFU setup for thermal treatment of cells. **b** Percentage of GFP positive (i.e., transcriptional activated by miniVPR) HEK cells expressing HSP-Cas, crTet, and pTRE3G-GFP, measured by flow cytometry 24 hr after cells were treated with HIFU. Heat treatment was done at 43 °C for 15 min. **c** Activation of all four genes measured by RT-qPCR after HIFU treatment at 43 °C for 15 min. **d** Percentage of GFP positive (i.e., edited by ABE8e) HEK cells expressing HSP-Cas, crGFP*, pCAG-GFP*, measured by flow cytometry 24 hr after cells were treated with HIFU. In **b**, **d** %GFP+ cells were calculated as the percentage of GFP-positive cells in the mCherry-positive population (i.e. cells that express the guide). Mean GFP values are shown in Supplementary Fig. 6. NT: non-treated. HIFU: high-intensity focused ultrasound-treated. TC: thermocycler. All data are shown for 2–5 independent replicates. Source data in this figure are provided as a Source Data file. An unpaired two-sided t-test was used for statistical analysis and p-values are presented in Supplementary Table 3.

## Discussion

In this project, we demonstrate the ability to control endogenous gene expression in vitro and in vivo using focused ultrasound. The smaller size of Cas12a (3.6 kb) compared to Cas9 (4.1 kb) facilitates easier delivery. We combined the recently developed hyperdCas12a and effectors with heat shock promoters and demonstrated the capability of our system to modulate multiple endogenous genes using a single sonogenetic stimulus. We are optimistic that this strategy can be extended to gene repression, epigenetic modifications, and many other modes of genetic engineering.

Ultrasound exhibits significant advantages over other chemical and physical stimuli, with deep penetration and spatio-temporal resolution. We choose to exploit the thermal effect of focused ultrasound as HIFU has already been used for tumor ablation in humans[41,42]. The precision of acoustic waves coupled with guided magnetic resonance or ultrasound imaging ensures on-target specificity and reduces off-target side effects. Though there have been similar approaches using ultrasound for promoter-driven gene expression[20–22,27], our approach incorporating the CRISPR-Cas system possesses the benefits of versatility and multiplexity. The small size of guide RNA and its vector, besides facile cellular delivery, significantly reduces cloning and engineering efforts. The guide array allows for the regulation of many genes simultaneously and array design optimization is an area of great interest. Several studies have implemented Boolean logic gates

by co-opting two CAR molecules, each carrying a different proximal T cell signaling molecule[4]; or using a SynNotch system to induce ROR1 CAR expression, thus activating T cells in the presence of two tumor antigens[5]. Our work is complementary to these studies by providing additional external controls by modulating endogenous genes in T cells to fine-tune their cellular behavior even after cell infusion.

In the current study, we detected a rapid downregulation of target gene expression (Supplementary Fig. 2a), implying that our ultrasound-induced transcriptional activation system induces short-term and reversible changes in gene expression. It is worth noting that the kinetics of activation and cessation of an endogenous gene upon stimulation can vary significantly depending on the cell type and the specific gene being targeted. The ultrasound stimulus offers the advantage of repeatable administration in a non-invasive manner, with precise control over its duration and amplitude. This suggests the potential for enhancing the initial upregulation of the target gene through subsequent rounds of ultrasound stimulation.

Our strategy can be applied in complex biological systems such as stem cell differentiation and T cell therapy where multiple genes are implicated and may possibly enhance their therapeutic effects. The main challenge to overcome is insufficient T cell activation due to the immunosuppressive tumor microenvironment and rapid exhaustion in solid tumors, in which several pathways and factors are involved[33–35]. There is a multitude of research focused on identifying new genes

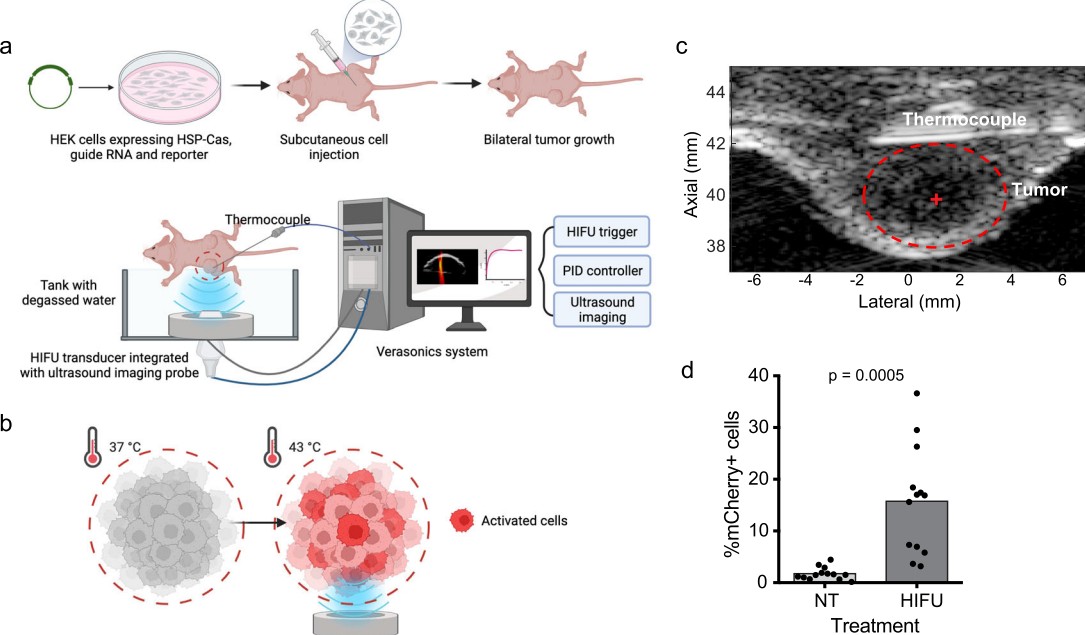

**Fig. 5 | HSP-hyperdCas12a is activated by HIFU in animal models. a** Schematic overview of ultrasound-activated hyperdCas12a for reporter gene expression in an animal model. HEK cells were transduced to stably expressing a pTRE3G-mCherry reporter, HSP-hyperdCas12a-miniVPR for transcriptional activation and guide RNA crTet targeting the TRE3G promoter. Engineered cells were subcutaneously injected into J:Nu mice for a bilateral tumor model. A dedicated ultrasound-guided focused ultrasound (USgFUS) setup combining imaging and therapy was utilized to target and treat the tumor. The HIFU treatment consisted of heating and maintaining a temperature of 43 °C for 15 min in one of the bilateral tumors. The tumor temperature was automatically regulated by the ultrasound PID controller with feedback from a needle thermocouple placed beneath the tumor. **b** Schematic showing expected mCherry reporter activation after ultrasound treatment. **c** Representative image acquired from ultrasound imaging probe, showing the tumor and needle thermocouple. HIFU transducer was focused to the center of the tumor marked with +. **d** Percentage of mCherry positive HEK cells measured by flow cytometry 24 hr after cells were treated with HIFU. %mCherry+ cells were calculated as the percentage of mCherry-positive cells in the GFP-positive population (i.e. cells that express the guide). NT: non-treated. HIFU: high-intensity focused ultrasound-treated at 43 °C for 15 min. Each data point represents one animal ($n = 13$, $p = 0.0005$). Source data in this figure are provided as a Source Data file. Paired two-sided t-test was used for statistical analysis and $p$-values are presented in Supplementary Table 3.

involved in these processes. For example, TRUCK CAR T cells rely on transgenic expression of relevant genes to increase CAR T efficiency[43]. Our preliminary data of heat-induced reporter and endogenous gene activation in primary human T cells suggest a more efficient and selective approach to target multiple genes and boost CAR-T efficiency. There will be no limit on the size of targeted genes and their vector delivery, because of the convenience of using small guide RNAs. Meanwhile, we observed a varied amount of reporter activation for temperatures ranging from 40 °C to 43 °C at different durations, suggesting that thermal-induced killing can not only be confined to a specific region of interest but is highly tunable to prevent side effects from non-specific and excessive activation.

We chose a sensitive promoter and mild hyperthermia conditions (43 °C for 15 min), which have been used by others conducting similar experiments for thermosensitive gene activation[20,36]. HIFU-stimulated hyperthermia (with temperature increases of a few degrees) is under study for the release of drugs from temperature-sensitive delivery vehicles and the temperature increase can be monitored and controlled in real-time[44–46]. HIFU ablation (achieving a temperature above 60 °C) is approved in the United States for the treatment of bone metastases, prostate cancer, uterine fibroids, and neurological diseases, and at sites around the world is approved for the treatment of breast, pancreatic, and liver tumors (https://cdn.fusfoundation.org/2023/07/21125203/FUSF-State-ofthe-Field-Report-2023_July-24.pdf)[47,48]. With thermal ablation, enhanced heat shock protein expression and antigen release are generally considered immunostimulatory and reduce tumor growth[49,50]. For immune stimulation, it is crucial to find optimal thermal/HIFU parameters to strike a balance between cell viability and activation effect.

To strengthen our strategy for in vivo applications, we need to improve ultrasound-mediated endogenous gene activation. We observed robust reporter gene activation in HIFU-treated cells but compromised performance in endogenous gene activation. The TRE3G promoter contains 7 copies of TetR binding sites, which the guide RNA can recognize and enable reporter gene expression; whereas for guides targeting endogenous genes, there is only one single binding site, thus limiting transcriptional activation. The extent of heat-induced gene activation also depends greatly on the efficiency of guide RNAs. We saw very little heat-induced activation for a gene with poor guides (i.e., limited gene activation even with a constitutively expressing hyperdCas12a-miniVPR). To resolve that, we can screen for more guide RNAs or use multiple guide RNAs for the same gene to increase gene activation. Besides better guide RNAs, we can also enhance hyperdCas12a expression through promoter engineering for increased heat-sensitivity. Further enhancing the sensitivity of our system is helpful in that a minimal amount of ultrasound stimulus will result in rapid expression of hyperdCas12a, which then utilizes a guide RNA for efficient endogenous gene activation.

The CRISPR toolbox is rapidly expanding for different modes of genome manipulation. The next step will be generalizing our ultrasonic control strategy to the expanding CRISPR-Cas toolbox for a series of Cas proteins including Cas9[51], RNA-guided RNA targeting Cas13[52], miniature CRISPR systems including Cas14[53], CasMINI[54] and their variants. Given the diversity and broad applications of CRISPR-Cas tools in areas like RNA-targeting[52], epigenetic modification[55], and transcriptional repression[56] as well as their rapid translation into clinical uses[57], we anticipate that ultrasound can be incorporated as an important safe and efficient physical stimulus to specifically monitor

these molecular changes when CRISPR-Cas is applied in pre-clinical and clinical models.

While this work exploits the thermal effect of focused ultrasound, its mechanical effect serves as another interesting physical stimulus. As abovementioned, ultrasound at different frequencies can activate many endogenous ion channels to trigger calcium influx and neuronal activation; but there has been limited work to connect ion channel activation with gene expression for in vivo applications. We hypothesize that we can also control CRISPR-Cas expression through a calcium-sensitive promoter (e.g., NFAT) where calcium influx induced by ultrasound can activate CRISPR-Cas and thus genetic engineering. Combined with the orthogonality of different types of CRISPR-Cas systems (e.g., transcriptional activation v.s. repression), it is possible to achieve multiplexed genetic engineering by linking different Cas expressions to either the thermal or mechanical effect of ultrasound. Given the variable parameters of ultrasound (e.g., frequency and intensity) that give rise to different outcomes (e.g., temperature and force), ultrasound has the potential to create a modular and versatile platform to accommodate the complexities and challenges of cell therapies.

## Methods

### Ethical statement
Animal studies were performed in accordance with the Guide for the Care and Use of Laboratory Animals of the National Institutes of Health, and all experiments were performed under a protocol approved by the Administrative Panel on Laboratory Animal Care (APLAC) and Institutional Animal Care and Use Committee (IACUC) of Stanford University. Primary human T cells were isolated from donor whole blood obtained from the Stanford Blood Center (Stanford, CA) and cultured in vitro under a protocol approved by the Stanford University Institutional Review Board.

### Plasmid cloning
Constructs were obtained by standard molecular cloning techniques and verified by sequencing. Fragments were assembled using InFusion (Takara Bio) and then transformed into Stellar Competent cells (Takara Bio). HyperdCas12a, miniVPR, and ABE8e sequences were amplified from plasmids in previous work[32]. Heat shock promoter was amplified from human genomic DNA (extracted from HEK293T cells) using the following primers (forward: gatctgaatggaatgttctggattgaaga, reverse: ggctgaagcttcttgtcggaa). Sequences of all the key constructs and guide RNA are presented in Supplementary table 1 and 2.

### Cell culture
Standard cell culture techniques were used and cells were maintained at 37 °C and 5% $CO_2$. Human embryonic kidney (HEK293T, Cat# CRL-3216) and Jurkat (Cat# TIB-152) were acquired from ATCC. HEK293T cells (wild type, TRE3G-GFP reporter, and transduced stable cells) were maintained in DMEM with high glucose, sodium pyruvate, GlutaMAX (Thermo Fisher) and supplemented with 10% FBS (Sigma), and 100 U/mL of penicillin and streptomycin (Gibco). HEK cells were passaged every 2-3 days before reaching confluence. Jurkat cells were maintained in RPMI 1640 (Thermo Fisher), supplemented with 10% FBS (Sigma) and 100 U/mL of penicillin and streptomycin (Gibco). Jurkat cells were seeded at $1 \times 10^5$ cells per mL and passaged every 3-4 days as the density approached $1 \times 10^6$ cells per mL. Primary human T cells were isolated and cultured in vitro under a protocol approved by the Stanford University Institutional Review Board. Briefly, T cells were isolated from donor whole blood obtained from the Stanford Blood Center (Stanford, CA) using the Dynabeads™ Human T-Activator CD3/CD28 for T Cell Expansion and Activation (Thermo Fisher). T cells were aliquoted and stored in liquid nitrogen. Thawed T cells were cultured in RPMI 1640 (Thermo Fisher), supplemented with 10% FBS (Sigma), 100 U/mL of penicillin and streptomycin (Gibco), and 200 U/mL rIL-2

(Thermo Fisher). T cells were stimulated with CD3/CD28 Dynabeads at a 1:1 cell:bead ratio (day 0). One day after bead activation, T cells were transduced with HSP-Cas lentivirus (day 1). On day 2, T cells were transduced with crRNA and TRE3G-GFP virus (for reporter activation) or crRNA (for endogenous gene activation). Dynabeads were removed on day 3 and T cells were maintained at $1 \times 10^6 - 2 \times 10^6$ cells per mL. On day 5, puromycin was added at 1 µg/mL to select for cells expressing HSP-Cas. Puromycin was removed on day 7 and replaced with fresh media containing rIL-2. Heat treatment was done on day 9 as described below and gene activation was assayed on day 10.

### Stable cell line production
Constructs were stably introduced to cells by lentiviral transduction. Lentiviruses were produced as previously reported[37]. Briefly, 9 million HEK293T cells were plated in 10 cm plate in 11 mL culture media. The following day, the transfection mix was prepared by adding 10.58 µg of construct of interest in a pHR vector, 7.56 µg of psPAX2, and 3.28 µg of pMD2g in 2.52 mL of OptiMEM reduced serum media (Thermo Fisher), followed by 64.26 µl of Mirus LT-1. The mixture was gently mixed and incubated at room temperature for approximately 20 min. Then 45% volume of culture media was removed from the HEK cells and the transfection mixture was added in a dropwise manner. About 6 hours later, all the media was aspirated and replaced with fresh culture media supplemented with ViralBoost reagent (Alstem, freshly added). The supernatant was collected 18 hours later and filtered through a 0.45-µm polyvinylidene fluoride filter (Millipore) and Lentivirus Precipitation Solution (Alstem) was added. The virus was precipitated at 4 °C for at least 4 hours to overnight and pelleted by centrifuging at x1500g for 30 min. The pellet was resuspended in 50 µL of OptiMEM, aliquoted in 10 µL each and stored at −80 °C. Transduction was done immediately after the cells were plated. For HEK cells, cells were plated $2 \times 10^5$ cells per well in 2 mL in a 6-well plate and one aliquot (10 µL) of virus was added dropwise to the cell media. Transduced cells were expanded to a 10 cm plate and selected with 1 µg/mL puromycin or selected for fluorescent marker expression. For Jurkat cells, cells are plated at $1 \times 10^5$ cells per well in 0.5 mL in a 24-well plate and one aliquot (10 µL) of virus was used per well. Transduced cells were expanded to a 6-well plate and selected with 2 µg/mL puromycin.

### Transfection
To test endogenous gene activation or base editing, HEK cells stably expressing HSP-hyperdCas12a were plated at ~75-100k per well in 0.5 mL media in a 24-well plate. Approximately 24 hr later, cells were transfected with crRNA plasmids using TransIT-LT1 transfecting reagent (Mirus). Briefly, to prepare the transfection mix for each well, 500 ng plasmid DNA was added to 50 µL OptiMEM, followed by 1.5 µL TransIT-LT1. The mixture was vortexed for mixing and incubated at room temperature for 15 min and added dropwise to cells. Transfection efficiency was usually more than 90%. Treatment was done 24 hr after transfection.

### Flow cytometry
Adherent cells were dissociated with 0.05% Trypsin-EDTA (Life Technologies) and PBS + 10% FBS was used to quench trypsin. Suspension cells were used directly in their growth media. Samples were measured in a CytoFLEX S flow cytometer (Beckman Coulter). Samples were run in a tube mode and for each sample, at least 10,000 cells expressing genes of interest (i.e., with fluorescent markers) were collected. For TRE3G-GFP reporter activation in HEK and T cells, cells were gated for mCherry + . For base editing in HEK and Jurkat cells, cells were gated for mCherry+ (i.e., guide-expressing). For CD2 and CXCR4 immunostaining in HEK cells, cells were gated for BFP+ (i.e., guide-expressing). To prepare tumor cells for flow cytometry analysis, the extracted tumors were placed in PBS in a 24-well plate. For each tumor, a fresh razor blade was used to cut the sample into small pieces in PBS. The

minced tissue and cells were transferred to a 1.5 mL tube and were further triturated by pipetting. The resulting cell suspension was passed through a 100 µm cell strainer into a new conical tube. Before analysis, the cell suspension was transferred to a flow cytometry tube with a cell strainer cap (Falcon®) to get a homogenous solution with no visible trunks of tissues/cells. The tumor cells were gated for GFP+ (i.e., guide-expression). Due to varied guide expression in tumors and loss of cells during sample preparation, we collected as many GFP+ cells as possible and at least 2000 cells were collected and analyzed. FlowJo v10 (BD Biosciences) was used for data analysis and compensation was applied. A sample gating strategy is provided in Supplementary Fig. 7.

## In vitro heat treatment
Adherent cells were dissociated with 0.05% Trypsin-EDTA (Life Technologies). Dissociated cells or suspension cells were pelleted by centrifuge. About $2 \times 10^5$ cells were resuspended in 30 µl of culture media in a 0.2 mL PCR tube. The cells were heated in a thermocycler (Bio-Rad C1000 Touch) at various temperatures and duration. After treatment, the cells were transferred to a plate with fresh culture media and maintained at 37 °C until analysis.

## ELISA
One day post-guide RNA transfection, heat treatment was conducted, and cells were transferred to their original culture media (i.e., media was preserved when cells were centrifuged). One day post-treatment, supernatant from cell cultures were collected and stored at −80 °C. Cytokines (IL2, IFN-γ, CXCL10) were detected by ELISA MAX Deluxe kits (BioLegend) in technical triplicate. A Synergy H1 plate reader (BioTek) and the BioTek Gen5 v.3.02 software were used to read absorbance at 450 nm and 570 nm. Cytokine concentration was calculated by fitting to a standard curve.

## Immunostaining
Cell were plated in a 24-well plate and guide transfection and heat treatment were done as abovementioned. One day after heat treatment, cells were harvested by adding 100 µL Accutase (STEMCELL technologies) and transferred to a 1.5 mL tube. Cells were centrifuged for 2 min to remove supernatant and then resuspended in 375 µL blocking buffer containing 10% FBS in PBS. Cells were centrifuged again to remove the blocking buffer and resuspended in 100 µL blocking buffer containing APC-CXCR4 (BioLegend, Clone 12G5, Cat # 306510, 1:500 dilution) or APC-CD2 (BioLegend, TS1/8, Cat # 309224, 1:50 dilution). Cells were incubated with antibodies (1 hr at R.T. for CXCR4 and 30 min 4 °C for CD2), protected from light. To remove the antibody and wash, cells were pelleted by centrifuging and resuspended in 375 µL blocking buffer. Cells were centrifuged again and resuspended in 150 µL blocking buffer and immediately analyzed by flow cytometry.

## RT-qPCR
HEK cells stably expressing HSP-HyperdCas12a were transfected with crRNA plasmids and treated with heat as described above. One day post heat treatment, total RNA was extracted using RNeasy Plus Mini Kit (QIAGEN) and stored at −80 °C. An iScript cDNA Synthesis kit (Bio-Rad) was used to obtain cDNA. qPCR reactions were done in 384 well plates with iTaq Universal SYBR Green Supermix (Bio-Rad) and run on a CFX384 Touch Real-Time PCR thermocycler (BioRad). All samples were run in technical triplicate. GAPDH expression was measured in all samples as a control. To quantify fold-change in gene expression, the ΔΔCq method was used.

## Hyperthermia setup
The ultrasound platform in the USgFUS setup was from Verasonics (Vantage 256, Kirkland, WA) and the small animal 1.5-MHz 128-element therapeutic array was from Imasonic (Voray sur l'Ognon, France). An imaging array (L12-5 38 mm, ATL) was used for image guidance and treatment planning. The hypodermic needle thermocouple (HYP-1) was from Omega Engineering Inc. (Stamford, CT). Acoustic coupling was ensured with degassed deionized water heated at 37 °C. In relation with the array geometry, the 8-foci generated by the HIFU array were evenly spread on an ellipse with major and minor axis lengths of 3.0 mm and 1.8 mm respectively. Additionally, the 8-foci pattern was constantly steered following an ellipse pattern with a 3.5 mm and 2.3 mm length for the major and minor axis respectively. The acoustic output was modulated by changing the duty cycle (from 0 % to 80 %) with a maximum intensity $I_{spta}$ of 20 W/cm² and a peak negative pressure of 540 kPa as measured with a calibrated hydrophone (HNP-0400; Onda, Sunnyvale, CA).

## In vivo experiment setup
Animal studies were performed in accordance with Guide for the Care and Use of Laboratory Animals of the National Institutes of Health, and all experiments were performed under a protocol approved by the Administrative Panel on Laboratory Animal Care (APLAC) and Institutional Animal Care and Use Committee (IACUC) of Stanford University. All mice used were female J:Nu homozygous mice (6-8weeks old, The Jackson Laboratory, Cat # 007850). Mice were kept at room temperature 20-24 °C with humidity 40-60% and 12 hour light/12 hour dark cycle. A total of 13 animals in three separate experiments were used. Engineered HEK293T cells were injected bilaterally into the flank of the mice. For each injection, there were approximately 10 million cells in 40 µl PBS and 40 µl Matrigel (Corning). When the tumor size reached 4-5 mm in about 7-10 days, we conducted HIFU treatment on one tumor while the other tumor remained as the untreated control. The needle thermocouple was inserted subcutaneously between the tumor and the body wall. Once the thermocouple was in place, the animal was positioned on its side with the tumor facing the therapeutic array. We conducted HIFU at 43 °C for 15 min. The animal body temperature was monitored with a rectal probe (RET-3, Physitemp Instruments, Clifton, NJ) throughout the experiment. About 24 hr after HIFU, both tumours were extracted for flow cytometry analysis.

## Data analysis and statistics
Data in the bar graphs are presented as individual points and the sample sizes are indicated in the figure legends. No randomization or blinding was performed. For statistical analysis in cultured cells (Figs. 1–4 and Supplementary Data Figs. 1–6), unpaired two-sided T tests were performed with Graphpad t test calculator. For Fig. 5, paired two-sided T-tests were performed with Graphpad t-test calculator. All the $p$-values are presented in Supplementary Table 3.

## Reporting summary
Further information on research design is available in the Nature Portfolio Reporting Summary linked to this article.

# Data availability
The authors declare that the source data supporting the findings of this study are provided with the manuscript, Supplementary Information, and Source Data file. Key constructs and plasmids will be available on Addgene (https://www.addgene.org/Stanley_Qi/). Source data are provided with this paper.

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

## Acknowledgements

The authors thank members of the Qi lab and the Ferrara lab for helpful discussions. We thank H. R. Kempton for providing the reporter construct for base editing. We thank V. Tieu and C. Chen for providing primary human T cells. Figures 1a, b, 4a, 5a, and 5b were prepared with Biorender.com with a purchased plan and permission to publish. This work is supported by the Stanford Bio-X Interdisciplinary Initiative Seed Grant. L.S.Q. acknowledges support from the National Science Foundation CAREER award (Award #2046650), National Institutes of Health (Grant # 1R01CA266470, 1R21AG077193, 1R21CA270609), and California Institute for Regenerative Medicine (CIRM, DISC2-12669). L.S.Q. is a Chan Zuckerberg Biohub – San Francisco investigator. K. W. F. acknowledges support from NIHR01CA253316 and R01CA112356.

## Author contributions

P.L. conceived the project, designed, and performed experiments, analyzed the data, prepared figures, and wrote the manuscript. J.F. set up the HIFU system, performed in vitro and in vivo HIFU treatment, and wrote the manuscript. Y.S. assisted in experiments for HEK and Jurkat gene activation and base editing and in vitro HIFU treatment. N.Z. assisted in experiments for HIFU treatment and prepared figures. A.J.K. assisted in tumor collection and flow cytometry. B.W. and M.N.R. maintained animals, performed subcutaneous cell injection, assisted in HIFU treatment, and collected tumors. L.S.Q. and K.W.F. conceived the project, provided supervision, prepared figures, and wrote the manuscript. All authors read and commented on the manuscript.

## Competing interests

L.S.Q. is a founder of Epic Bio and a scientific advisor of Kytopen and Laboratory of Genomics Research. The remaining authors declare no competing interests.
