## [Peer Review File · Nature Communications]

Reviewers' Comments:

Reviewer #1:

Remarks to the Author:

The manuscript by Liu et al describes the use of engineered heat shock promoters and a nuclease-dead Cas to thermally control the expression of endogenous genes in HEK cells, Jurkat cells and primary T-cells. Most of the experiments are performed in vitro with heating provided by a thermocycler. In addition, there is an in vitro demonstration and an in vivo demonstration (with subcutaneously implanted HEK cells) where heating is performed by focused ultrasound (HIFU). Overall, this is a high-quality study with convincing data.

The primary novelty claimed relative to previous literature on HSP-driven genetic control is that this work introduces a way to regulate endogenous genes with a Cas-based system. However, a previous study by Gamboa et al (<https://doi.org/10.1021/acscchembio.9b01005>) already demonstrated HSP-driven expression of dCas9 for both activating and repressing endogenous genes selected with guide RNAs. This previous study demonstrated in vitro and in vivo control, and the in vivo experiments also used subcutaneous HEK cells. While the hyperthermia technique differs between that study (infrared) vs this one (HIFU), I don't think this difference itself is very significant since heat is heat (for the most part) and HIFU hyperthermia is well-established for transcriptional activation. Parenthetically, I find it somewhat surprising that the authors did not cite this previous paper given its direct relevance and their citation of several other studies from the same group.

In light of the literature, I think the novel aspects of Liu et al's study are (1) the use of a smaller and more efficient Cas, (2) the multiplexed regulation of several genes at the same time rather than a single one, and (3) base editing (shown with an engineered reporter). I am not sure if these advances on their own make the manuscript most appropriate for Nature Comms vs a more specialized (e.g. synthetic biology) journal. Also, previous papers in this field published in similar journals had more advanced and translationally relevant in vivo demonstrations (e.g. enhanced solid tumor control by heat-activated cells in refs 16, 17 and 27). However, I would defer to the editor if they think the novelty and extent of demonstration in this manuscript are suitable for their journal.

Technically, the paper is sound and I have no major comments. One minor note is that the results for HIFU activation and editing in Fig 3 are substantially weaker than for contact heating, which appears to suggest that ultrasound does a worse job of activating cells at a given temperature. However, there is uncertainty about the actual temperature reached by the cells. Why not measure the temperature directly in the well containing the cells rather than outside the tube to remove ambiguity and either reinforce the conclusion about HIFU being different or show that HIFU and contact heating produce the same results?

Reviewer #2:

Remarks to the Author:

The authors have created an ultrasound-responsive system to enable spatiotemporal control of gene activation in vitro and in vivo. This work is an extension of previous publications that have utilized heat-responsive promoters to control gene expression, but adds the potential for facile multiplexed gene editing or regulation through base editing or CRISPRa.

I have several major concerns:

1. The authors don't adequately demonstrate how the technology (i.e., placing either a base editor or CRISPRa transgene under the control of a heat-responsive promoter) described by this manuscript enables any applications that aren't already accessible through existing technologies. I agree that the power of the system is in multiplexed gene editing or CRISPRa (as laid out in Figure 1a), but the authors only use this technology to target, at most, two genes simultaneously, and in most applications they only target a single gene. These applications could have been enabled by simply putting the gene(s) under the control of the heat shock promoter.

a. I recommend performing experiments that use higher-order multiplexing to convincingly demonstrate the utility of the system over existing technologies in vitro and/or in vivo.

2. Line 205: the authors suggest that the spatiotemporal control of gene expression displayed by their system could enable CAR Ts with improved on-target/off-tumor antigen targeting capabilities, presumably through limiting CAR expression to T cells in a tumor (though this wasn't clear in the text). However, the authors fail to demonstrate that:

a. Primary human T cells that employ their gene activation system to selectively activate CAR expression do not express sufficient levels of CAR protein at 37C to enable those cells to respond to antigen expression in healthy tissue. The authors should demonstrate experimentally that their system does in fact mitigate the potential for on-target/off-tumor CAR T reactivity in vitro and/or in vivo.

b. T cells are capable of trafficking into and out of tumors, and activation of CAR expression in a tumor could result in CAR-expressing T cells in healthy tissues following trafficking out of the site of induction. Please provide data supporting the kinetics of protein expression following induction and cessation of gene activation and a discussion supporting the advantages and limitations of this system, including comparisons to other leading CAR T regulatory systems in the field (e.g., Tousley et al 2023 and Srivastava et al 2019).

c. Spatiotemporal control of CAR expression could be achieved by simply putting the CAR transgene under the control of the heat shock promoter, as noted by the authors (reference 16; Wu et al, 2021). The authors should demonstrate that the multiplexability of their system could enable higher-order control of gene expression that could have CAR T applications above and beyond existing technologies.

Minor concerns:

1. Line 31: what specific evidence exists that the precise regulation of genes at the site of disease results in improved treatment efficiency and safety? Please provide references.

2. Lines 371-382 should be rephrased to emphasize their speculative nature regarding the potential to create mechano-sensitive systems for programmable gene activation.

Reviewer #3:

Remarks to the Author:

Ultrasound-mediated sonogenetic control of genome regulation and base editing

In this manuscript the authors present a novel strategy for the regulation of endogenous gene expression through the coupling of thermogenetic transcriptional control and CRISPR technologies. Utilizing a previously described, hyperdCas12a-miniVPR effector under the control of a heat shock promoter, the authors are able to transcriptionally activate the expression of a GFP reporter in a temperature dependent manner. The authors subsequently demonstrate the activation of several endogenous target genes in multiple cellular contexts and using dual gRNA arrays. In addition to CRISPRa, the authors also explore thermoregulation in the context of a base-editing reporter. Finally, the authors demonstrate the use of ultrasound as a strategy for thermal induction of CRISPRa in both in vitro cell culture and an in vivo murine implantation model.

The challenge of achieving robust, precise, non-invasive gene regulation is critical for manipulating complex cellular models and for the safety and efficacy of cell-based therapeutics. The authors of this manuscript demonstrate a creative approach towards these ends and this work provides a compelling proof-of-concept for future platform development. While conceptually intriguing we feel that this work, as presented, is underdeveloped for publication in Nature Communications without expanded studies.

Broad critiques:

Many of the potential research and clinical applications for thermally activated gene expression may necessitate sustained expression of a target gene. Unlike gene editing strategies which only

require acute expression for permanent genome modification, robust CRISPR mediated transcriptional activation will likely necessitate the sustained expression of CRISPRa effector proteins to maintain stable activation. To enhance their manuscript, the authors should provide data demonstrating how long after thermal treatment gene activation is sustained. Furthermore, some endogenous genes may require days or weeks to activate if chromatin remodeling is required. The authors should clarify why they decided to perform most of their experiments at 24hrs and provide a longer time-course of viability, target activation and subsequent downregulation for a subset of endogenous genes. The authors could consider exploring the effect of pulsing thermal/ultrasound treatment for longer term activation.

While the engineering of a stable system may be required for sustained CRISPRa, conversely, one challenge with stably engineered inducible systems is the issue of unintended gene activation or leakiness. This could additionally be problematic in the context of permanent genome modification (such as with the base-editor) where leaky editing could result in the accumulation of edits over time. The authors should consider exploring how tight the regulation of their base-editor is over time with sustained culture. They could consider benchmarking this against established inducible systems like tet or cumate.

For the TRE3G-fluorescent reporter the authors should clarify the number of predicted crRNA binding sites in the text and consider including a schematic in the supplement. Assuming the authors used the full length 7X binding promoter, do the authors believe that this explains the relative efficacy of the reporter activation vs endogenous gene activation? The authors attempted to activate multiple genes with a dual gRNA tandem array but have the authors explored multiple gRNAs for the same gene as a strategy to enhance activation?

Thermal exposure in cells will likely activate endogenous heat-shock response pathways. The authors should discuss how this may influence the application of this technology. For all main figures the authors should include non-targeting gRNA controls to demonstrate that target gene activation is specifically the result of gRNA targeting and not heat exposure (e.g. Fig 1d,e; Fig2b.; Fig 3).

Overall, the efficacy of the ultrasound mediated gene activation seems moderate both when compared directly with thermal activation in vitro as well as induction of the CRISPRa reporter in vivo. Practically this could limit the application of this technology more broadly without further optimization. Since the use of ultrasound in this manuscript is somewhat limited the authors should remove it from their title or demonstrate further optimization of this technology. Likewise, the use of base editing is limited to targeting of a reporter in vitro. The authors should remove this from their title or expand their experiments to include the editing and analysis of endogenous loci.

Viability following thermal exposure is clearly a challenge but it is unclear how cell viability is impacted by ultrasound treatment. The authors should clearly evaluate this in their in vitro model.

Specific points:

To increase the accessibility and clarity of this work please provide sequence information for critical vectors as .gb or annotated word files.

- In particular, the primers listed in the methods section for heat shock promoter cloning could be predicted to amplify both the HSPA6 promoter and upstream of the similar HSPA7 pseudogene. A full vector sequence would clarify which sequence was used and enable these studies to be reproduced more easily.
- As mentioned previously, the sequence of the TRE3G-fluorescent reporter would be useful to clarify how many predicted crRNA target sites are retained in the promoter.

Appropriate statistical tests should be added throughout and indicated within the figure and/or figure legends.

Where possible for flow cytometry data it would be helpful to include both % positive and MFI information.

Figure 1b very usefully specifies the method of expression (viral vs stable integration) in the schematic. The authors should do this throughout the manuscript.

In the description of the multiplex gRNA experiments (lines 142/143) the text reads "one guide for INFG and 2 guides for IL2". This appears to be inconsistent with the schematic in Fig 1e which shows 1 gRNA for each gene (2 gRNAs total). The authors should clarify their approach.

For similar assays the authors should be consistent in the labeling of their graphs as Mean fluorescence or protein expression A.U.

For clarity the authors should make clear in the text and/or figure legends how experimental data is normalized.

For supplemental 1b the references to the left/right panels in the legend seem to be reversed.

Overall summary of the revision

A summary of major revisions is as follows:

- We have characterized the ability of our system to simultaneously activate multiple endogenous genes under the control of heat treatment (**Fig. 2b**) or ultrasound (**Fig. 4c**). We demonstrated that the system could efficiently control up to four endogenous genes, a capability not demonstrated previously.
- We carried out detailed kinetic characterization of our system to quantify the duration of target gene activation after stimulus retrieval (**Extended Data Fig. 2a**), concluding rapid induction and temporary expression of target genes.
- We performed a long time-course experiment, over three weeks, to characterize basal HSP-induced hyperdCas12a-miniVPR expression without heat treatment (**Extended Data Fig. 2b**), concluding that our system showed almost no basal leakage expression of the effector molecule.
- We've revised the main text to add new citations and address reviewers' questions.

REVIEWER COMMENTS

Reviewer #1 (Remarks to the Author):

The manuscript by Liu et al describes the use of engineered heat shock promoters and a nuclease-dead Cas to thermally control the expression of endogenous genes in HEK cells, Jurkat cells and primary T-cells. Most of the experiments are performed in vitro with heating provided by a thermocycler. In addition, there is an in vitro demonstration and an in vivo demonstration (with subcutaneously implanted HEK cells) where heating is performed by focused ultrasound (HIFU). Overall, this is a high-quality study with convincing data.

We thank the reviewer for the constructive comments on our manuscript.

The primary novelty claimed relative to previous literature on HSP-driven genetic control is that this work introduces a way to regulate endogenous genes with a Cas-based system. However, a previous study by Gamboa et al (<https://doi.org/10.1021/acscchembio.9b01005>) already demonstrated HSP-driven expression of dCas9 for both activating and repressing endogenous genes selected with guide RNAs. This previous study demonstrated in vitro and in vivo control, and the in vivo experiments also used subcutaneous HEK cells. While the hyperthermia technique differs between that study (infrared) vs this one (HIFU), I don't think this difference itself is very significant since heat is heat (for the most part) and HIFU hyperthermia is well-established for transcriptional activation. Parenthetically, I find it somewhat surprising that the authors did not cite this previous paper given its direct relevance and their citation of several other studies from the same group.

The Gamboa *et al.* engineered and tested a thermosensitive dCas9-VP64 and KRAB-dCas9 and demonstrated *in vivo* KRAB-dCas9 activation by NIR and implanted gold nanorod for transcriptional repression. Compared to photothermal-mediated control, our work has the following differences and advantages which make our work unique and significant:

- 1) We demonstrated multiplexed endogenous gene activation (we achieved 4 genes simultaneously in our new experiments), using the unique capability of the Cas12a system to process a compact guide RNA array for multiplexed gene targeting. Furthermore, we demonstrated thermal-controlled base editing. Both features are important but have not been demonstrated with the previous dCas9 system.
- 2) We showed preliminary data of our system in primary cells, which is recognized to be challenging for engineering with CRISPR systems.
- 3) Our use of ultrasound is non-invasive and does not require additional materials (e.g., implanted gold nanorod in the above study) to introduce photothermal energy. Ultrasound is therefore more penetrative and clinically relevant for cell therapies.
- 4) While Gamboa and co-workers showed *in vivo* gene suppression, we worked on *in vivo* gene activation, which supplements their strategy.
- 5) For their *in vivo* study, they increased local temperature to 44°C for 30 min and mentioned in figure legend (Figure 4b) that “This temperature was chosen to compensate for lower temperatures at the core of the implant due to NIR-light scattering by tissue and heat diffusion”, which is due to the scattering nature of light. In our case, we used 43°C for 15 min and observed robust transcriptional activation.

To address the reviewer comment, we have added the following discussion and citation in the main text line 64-72:

“One group has recently engineered HSP-mediated thermosensitive dCas9 with effectors and demonstrated *in vivo* transcriptional repression of individual genes by KRAB-dCas9 using near-infrared (NIR) excitation and implanted gold nanorod¹⁸. Compared to this work, our strategy uses the Cas12a family of proteins for multiplexed gene activation or base editing via ultrasound-mediated control, which is more penetrative and clinically relevant compared to NIR. Furthermore, our approach is more scalable, as any one or more genes of interest can be flexibly targeted by a compact guide RNA array, which can be conveniently synthesized for easy delivery compared to the traditional promoter-based cloning (i.e., each gene requires a different promoter).”

In light of the literature, I think the novel aspects of Liu et al’s study are (1) the use of a smaller and more efficient Cas, (2) the multiplexed regulation of several genes at the same time rather than a single one, and (3) base editing (shown with an engineered reporter). I am not sure if these advances on their own make the manuscript most appropriate for Nature Comms vs a more specialized (e.g. synthetic biology) journal. Also, previous papers in this field published in similar journals had more advanced and translationally relevant *in vivo* demonstrations (e.g. enhanced solid tumor control by heat-activated cells in refs 16, 17 and 27). However, I would defer to the editor if they think the novelty and extent of demonstration in this manuscript are suitable for their journal.

Thank you for recognizing the advantages of the Cas12a system in this work. We believe the ability for multiplexed gene regulation in primary cells under the ultrasound control is an important advance. In this work, we combined ultrasound with CRISPR-Cas tools for transcriptional activation and base editing with proof-of-principle demonstrations in various cell types (including primary T cells) and a mouse model. We believe that our strategy has important impacts in the following areas:

- 1) There have been few papers on ultrasound-mediated endogenous gene expression. Most works cloned the gene of interest directly under the heat shock promoter for ectopic expression. When the gene of interest has a large size, it will be difficult for delivery. The same problem exacerbates for multiple genes. It is practically challenging to deliver multiple genes in primary cells or *in vivo* and still warranty their expression under the control of ultrasound. On the other hand, our method of using a compact guide array,

which is smaller in size (~40bp per guide RNA, thus a 5-guide array is only ~200 bp), not only allows us to target any gene with ease but also offers the possibility to target multiple genes simultaneously. In our revised manuscript, we have presented data on simultaneous activation of four endogenous genes using heat or focused ultrasound.

- 2) In T cell engineering, delivery and expression of large vectors have been a technical challenge. In our work, we have successfully demonstrated that HSP-hyperdCas12a-miniVPR and guide RNA work with a commonly used lentivirus-based transduction protocol in T cells. T cells can express hyperdCas12a-miniVPR after thermal activation and different genes can be efficiently activated (GFP v.s. FOXQ1) with guide RNAs.
- 3) With data on base-editing, we presented that the modularity of the Cas12a platform enables different modes of genome engineering by switching the effectors.

In summary, we believe our work is suitable for Nature Communications and targets a broad scientific community. Researchers from different fields may find our work interesting, including synthetic biology, ultrasound therapeutics, immunotherapy, cell therapy, and genome engineering.

Technically, the paper is sound and I have no major comments. One minor note is that the results for HIFU activation and editing in Fig 3 are substantially weaker than for contact heating, which appears to suggest that ultrasound does a worse job of activating cells at a given temperature. However, there is uncertainty about the actual temperature reached by the cells. Why not measure the temperature directly in the well containing the cells rather than outside the tube to remove ambiguity and either reinforce the conclusion about HIFU being different or show that HIFU and contact heating produce the same results?

To measure temperature inside the tube, we insert the needle thermocouple inside the cell suspension and would risk contamination in the full studies since we continue to culture the cells for another 24 hr before analysis. Due to this concern, in mock experiments, we validated that the temperature was accurately assessed by our method. Also, with a thermocouple inside the tube, it will generate an artifact.

The cells sink to the bottom of the tube during treatment, so measuring the temperature immediately next to the tip of the tube is an accurate reflection of the temperature inside the tube. There is also heat transfer between the outside/phantom and inside of the tube, thus the temperature is equilibrated.

Reviewer #2 (Remarks to the Author):

The authors have created an ultrasound-responsive system to enable spatiotemporal control of gene activation *in vitro* and *in vivo*. This work is an extension of previous publications that have utilized heat-responsive promoters to control gene expression, but adds the potential for facile multiplexed gene editing or regulation through base editing or CRISPRa.

I have several major concerns:

1. The authors don't adequately demonstrate how the technology (i.e., placing either a base editor or CRISPRa transgene under the control of a heat-responsive promoter) described by this manuscript enables any applications that aren't already accessible through existing technologies. I agree that the power of the system is in multiplexed gene editing or CRISPRa (as laid out in Figure 1a), but the authors only use this technology to target, at most, two genes simultaneously, and in most applications they only target a single gene. These applications could have been enabled by simply putting the gene(s) under the control of the heat shock promoter.
 - a. I recommend performing experiments that use higher-order multiplexing to convincingly demonstrate the utility of the system over existing technologies *in vitro* and/or *in vivo*.

Thank you for this suggestion. We have now tested a 5-guide array targeting 4 endogenous genes, including IL18, IL7, IFN γ and IL2, with our HSP-Cas system under heat or ultrasound conditions. We transfected the 5-guide array into HEK293T cells stably expressing HSP-hyperdCas12a-miniVPR, exposed the cells to elevated temperatures (43°C for 15 min) in a thermocycler or HIFU, and quantified mRNA level by RT-qPCR 24 hours after treatment. We were able to observe increased expression of all four genes (tens to hundreds of fold of activation) upon thermocycler or HIFU treatment (**Figure 2b, Figure 4c**). To our knowledge, we are not aware of ultrasound-mediated multiplexed endogenous gene control, which is an important feature to enhance engineered cells when applied to cell therapies.

To address the reviewer comments, we added the new data (**Figure 2b, 4c**) and the following description in the main text:

Line 157-164: "We next generated a gRNA array containing five tandem guides targeting four endogenous genes, including *IFNG*, *IL2*, *IL18* and *IL7* (**Fig. 2b**). To ensure efficient gRNA processing, we used an optimized repeat sequence in the gRNA array²⁵. After transfecting the gRNA array and conducting heat shock in a thermocycler, we extracted mRNA and quantified expression of these genes by RT-qPCR. We observed a significant increase in the expression (tens to hundreds of fold of activation) for all four genes, compared to the non-treated samples, suggesting that the expression of multiple genes can be modulated simultaneously with one single stimulus (**Fig. 2b**)."

Fig. 2: Heat-induced hyperdCas12a for multiplexed gene activation and base editing in HEK cells. b, Top: construct of gRNA array containing 5 guides targeting IL18, IL7, IFN γ and IL2. Bottom: simultaneous activation of four genes measured by RT-qPCR after heat treatment at 43°C for 15 min.

Line 291-293: “We next tested the gRNA array targeting four endogenous genes in HIFU-treated cells. We observed substantial expression of *IFNG*, *IL2*, *IL18* and *IL7* in HIFU-treated cells, with 4.5-, 8.8-, 3.6-, 44.0- fold increase respectively, compared to the untreated cells (**Fig. 4c**).”

Fig. 4. High intensity focused ultrasound triggers HSP-hyperdCas12a activities for multiplexed gene activation and base editing in vitro. c, Activation of all four genes measured by RT-qPCR after HIFU treatment at 43°C for 15 min.

2. Line 205: the authors suggest that the spatiotemporal control of gene expression displayed by their system could enable CAR Ts with improved on-target/off-tumor antigen targeting capabilities, presumably through limiting CAR expression to T cells in a tumor (though this wasn't clear in the text). However, the authors fail to demonstrate that: a. Primary human T cells that employ their gene activation system to selectively activate CAR expression do not express sufficient levels of CAR protein at 37C to enable those cells to respond to antigen expression in healthy tissue. The authors should demonstrate experimentally that their system does in fact mitigate the potential for on-target/off-tumor CAR T reactivity in vitro and/or in vivo.

We'd like to clarify a few aspects that may have caused confusion. The focus of this paper is a proof-of-principle demonstration that ultrasound can be combined with cutting-edge CRISPR-Cas

tools for multiplexed endogenous gene expression in various biological contexts including cell lines, primary cells and in animal models. As a non-invasive and precise tool, ultrasound can be readily utilized for manipulating cell biology and function in a highly specific manner. It is not our intention to focus on CAR T cells, because a previous work has demonstrated inducible ectopic CAR activation using ultrasound-activated HSP promoter (Wu et al, 2021). The novelty and advantages of our system lie on its ability for multiplexed gene control and modularity (e.g. other modes of genetic engineering besides transcriptional activation) through the CRISPR-Cas tools. In this work, we have provided preliminary data that heat-activated hyperdCas12a-miniVRP worked in primary T cells to enhance a reporter and endogenous gene activation. We believe that demonstrating the current system can mitigate the on-target/off-tumor CAR T reactivity *in vitro* and/or *in vivo* is out of scope, because this mitigation is highly dependent on the T cell biology and the genes of choice. As a long-term goal, we are interested in using ultrasound to activate a panel of endogenous genes that are biologically important in T cell's therapeutic function (e.g., proliferation, longevity, cytotoxicity, reduced exhaustion) to improve CAR or TCR-T activity. We also anticipate applications of our technology in other biological contexts such as cellular reprogramming where many genes are involved, which can be readily achieved with our multiplexed gene control platform.

We have modified the main text in line 201-212:

“A major hurdle for both CAR-T cells and for native T cells is insufficient T cell activation due to a paucity and heterogeneity of tumor antigen expression, immunosuppressive tumor microenvironment, and rapid T cell exhaustion²⁷⁻²⁹. On the other hand, non-specific immune cell activation causes on-target, off-tumor adverse effects, leading to severe problems like neurotoxicity and cytokine release syndrome. Thus, there is a high demand for the development of more precise, non-invasive, and tunable control of engineered immune cells *in vivo*. Ultrasonic control of endogenous gene expression in T cells after cell infusion into the body offers new opportunities to manipulate cell activity with high spatial and temporal precision at the local sites of solid tumors¹⁵. We hypothesize that our strategy has the potential to enhance T cell activity through a heat-induced CRISPR-Cas system both in the context of CAR-T cells and in the context of wild type T cell receptors.”

b. T cells are capable of trafficking into and out of tumors, and activation of CAR expression in a tumor could result in CAR-expressing T cells in healthy tissues following trafficking out of the site of induction. Please provide data supporting the kinetics of protein expression following induction and cessation of gene activation and a discussion supporting the advantages and limitations of this system, including comparisons to other leading CAR T regulatory systems in the field (e.g., Tousley et al 2023 and Srivastava et al 2019).

Following the suggestion, we have conducted new kinetics experiments to look at the time-dependent activation and cessation of a GFP reporter gene. We used HEK293T cells stably expressing TRE3G-GFP and hU6-crTet-HSP-hyperdCas12a-miniVPR-CMV-mCherry. We heated the cells and measured mean GFP fluorescence intensity and percentage of GFP positive cells at different time points after heat activation until the cells reached confluency in the culturing plate. We began to see GFP activation in just 8 hours after heat induction. Activation reached maximum around 24 hours after the treatment and declined to baseline at 80 hours. Our strategy using hyperdCas12a-miniVPR controls the transcription of mRNA, which is temporary manipulation of gene expression without any permanent change to the genome. Furthermore, the temperature and duration can be precisely controlled to induce different extent of activation (such as a lower fold of activation), making this strategy modular and tunable.

We have added the following discussion in the main text line 111-116:

“We conducted kinetics experiments in HEK293T TRE3G-GFP reporter cells to monitor the time-dependent activation and cessation using the HSP-hyperdCas12a-miniVPR system. We heated the cells at 43°C for 15 min and characterized GFP activation at different time points until the cells reached confluency. We observed rapid GFP activation in 8 hr after heat induction. Gene activation reached maximum after 24 to 30 hr and started to decline to close to baseline at 80 hr after the treatment (**Extended Data Fig. 2a**).”

Extended Data Fig. 2: Characterization of time-course heat-induced hyperdCas12a-mediated gene activation in HEK293T cells. a, Left: mean GFP fluorescence intensity measured by flow cytometry at different time points after cells were treated at 43°C for 15 min. Right: percentage of GFP positive cells measured at different time points.

Most existing technologies for CAR T systems (e.g., Tousley et al 2023 and Srivastava et al 2019) use a Boolean logic gate approach whereby expression of CAR and activation of T cells depends on antigen-specific expression on tumor cells but not on healthy tissues. For example, in Tousley 2023, they implemented a Boolean logic gate by co-expressing two CAR molecules, each carrying a different proximal T cell signaling molecule. In Srivastava et al 2019, they built a SynNotch system to induce ROR1 CAR expression, thus forming a sequential cascade that can activate T cells in the presence of both tumor antigens. While these works nicely exploit the intracellular components to enhance tumor recognition specificity, our work is complementary to these studies by providing additional external controls on T cells. In that sense, once the cells are engineered ex vivo and injected back to patients, our method uses a non-invasive physical stimulus which allows us to control multiple endogenous genes in these T cells to finetune their cellular behavior even after cell infusion. We believe this method complements logic-gated expression and would potentially further improve specificity against tumor cells. We have cited these works in the paper and added the following sentence to describe it in Discussion line 372-377:

“Several studies have implemented Boolean logic gates by co-opting two CAR molecules, each carrying a different proximal T cell signaling molecule⁵²; or using a SynNotch system to induce ROR1 CAR expression, thus activating T cells in the presence of two tumor antigens⁵⁷. Our work is complementary to these studies by providing additional external controls by modulating endogenous genes in T cells to finetune their cellular behavior even after cell infusion.”

c. Spatiotemporal control of CAR expression could be achieved by simply putting the CAR transgene under the control of the heat shock promoter, as noted by the authors (reference 16; Wu et al, 2021). The authors should demonstrate that the multiplexability of their system could enable higher-order control of gene expression that could have CAR T applications above and beyond existing technologies.

To address this comment, as abovementioned, we constructed a five-guide array targeting four endogenous genes and demonstrated simultaneous activation of all the genes upon thermal treatment either in a thermocycler or focused ultrasound. We purposely designed an array

targeting genes of proinflammatory cytokines, and we foresee great potential of our strategy in cytokine therapy. Cytokine therapy as a single reagent often has limited efficacy in immunotherapy due to toxicity from systemic administration. Thus, controlled release at a defined timepoint and location restricts the effect of cytokines on targeted tumor only. Besides cytokines, we can also target other genes of interest by incorporating specific guide RNAs in the array design. We can target CXCL10 which promotes recruitment of other immune cells or IL12 which increases T cell cytotoxicity or cJun which prevents exhaustion. Immune response and T cell functionality are modulated by a series of cytokines and factors, which will be hard to change with the traditional approach. The ability for multiplexed gene regulation will be crucial to control cell activity more efficiently. For example, upregulation of a panel of genes using a guide array will collectively enhance function and anti-tumor activity of T cells from proliferation, longevity, and cytotoxicity. These proposed experiments are undergoing in the lab to study the specific biological impact of our method. This current paper emphasizes more on technology development and proof-of-concept demonstration. We hope to provide a versatile genetic engineering tool for scientists in the field of immunotherapy and cellular engineering/programming with the ability and specificity of modulating several genes simultaneously.

To address the reviewer comments, we added the new data (**Figure 2b, 4c**) and the following description in the main text:

Line 157-164: “We next generated a gRNA array containing five tandem guides targeting four endogenous genes, including *IFNG*, *IL2*, *IL18* and *IL7* (**Fig. 2b**). To ensure efficient gRNA processing, we used an optimized repeat sequence in the gRNA array²⁵. After transfecting the gRNA array and conducting heat shock in a thermocycler, we extracted mRNA and quantified expression of these genes by RT-qPCR. We observed a significant increase in the expression (tens to hundreds of fold of activation) for all four genes, compared to the non-treated samples, suggesting that the expression of multiple genes can be modulated simultaneously with one single stimulus (**Fig. 2b**).”

Fig. 2: Heat-induced hyperdCas12a for multiplexed gene activation and base editing in HEK cells. b, Top: construct of gRNA array containing 5 guides targeting IL18, IL7, IFNγ and IL2. Bottom: simultaneous activation of four genes measured by RT-qPCR after heat treatment at 43°C for 15 min.

Line 291-293: “We next tested the gRNA array targeting four endogenous genes in HIFU-treated cells. We observed substantial expression of *IFNG*, *IL2*, *IL18* and *IL7* in HIFU-treated cells, with 4.5-, 8.8-, 3.6-, 44.0- fold increase respectively, compared to the untreated cells (**Fig. 4c**).”

Fig. 4. High intensity focused ultrasound triggers HSP-hyperdCas12a activities for multiplexed gene activation and base editing in vitro. c, Activation of all four genes measured by RT-qPCR after HIFU treatment at 43°C for 15 min.

Minor concerns:

1. Line 31: what specific evidence exists that the precise regulation of genes at the site of disease results in improved treatment efficiency and safety? Please provide references.

We provide the relevant references to Boolean logic-gated expression:

49. Kloss, C. C. et al. Combinatorial antigen recognition with balanced signaling promotes selective tumor eradication by engineered T cells. *Nature biotechnology* **31**, 71-75 (2013).
50. Roybal, K. T. et al. Engineering T cells with customized therapeutic response programs using synthetic notch receptors. *Cell* **167**, 419-432 (2016).
51. Fedorov, V. D. et al. PD-1–and CTLA-4–based inhibitory chimeric antigen receptors (iCARs) divert off-target immunotherapy responses. *Science translational medicine* **5**, 215ra172-215ra172 (2013)
52. Tousley, A. M. et al. Co-opting signalling molecules enables logic-gated control of CAR T cells. *Nature* **61**, 507-516 (2023).
57. Srivastava, S. et al. "Logic-gated ROR1 chimeric antigen receptor expression rescues T cell mediated toxicity to normal tissues and enables selective tumor targeting." *Cancer cell* **35**, 489-503 (2019).

2. Lines 371-382 should be rephrased to emphasize their speculative nature regarding the potential to create mechano-sensitive systems for programmable gene activation.

Thank you for the suggestion. We have changed the text in line 435-443 to:

"We hypothesize that we can also control CRISPR-Cas expression through a calcium-sensitive promoter (e.g., NFAT) where calcium influx induced by ultrasound can activate CRISPR-Cas and thus genetic engineering. Combined with the orthogonality of different types of CRISPR-Cas systems (e.g., transcriptional activation v.s. repression), it is possible to achieve multiplexed genetic engineering by linking different Cas expression to either the thermal or mechanical effect of ultrasound. Given the variable parameters of ultrasound (e.g., frequency and intensity) that give rise to different outcomes (e.g., temperature and force), ultrasound has the potential to create a modular and versatile platform to accommodate the complexities and challenges of cell therapies.

Reviewer #3 (Remarks to the Author):

Ultrasound-mediated sonogenetic control of genome regulation and base editing

In this manuscript the authors present a novel strategy for the regulation of endogenous gene expression through the coupling of thermogenetic transcriptional control and CRISPR technologies. Utilizing a previously described, hyperdCas12a-miniVPR effector under the control of a heat shock promoter, the authors are able to transcriptionally activate the expression of a GFP reporter in a temperature dependent manner. The authors subsequently demonstrate the activation of several endogenous target genes in multiple cellular contexts and using dual gRNA arrays. In addition to CRISPRa, the authors also explore thermoregulation in the context of a base-editing reporter. Finally, the authors demonstrate the use of ultrasound as a strategy for thermal induction of CRISPRa in both in vitro cell culture and an in vivo murine implantation model.

The challenge of achieving robust, precise, non-invasive gene regulation is critical for manipulating complex cellular models and for the safety and efficacy of cell-based therapeutics. The authors of this manuscript demonstrate a creative approach towards these ends and this work provides a compelling proof-of-concept for future platform development. While conceptually intriguing we feel that this work, as presented, is underdeveloped for publication in Nature Communications without expanded studies.

Broad critiques:

Many of the potential research and clinical applications for thermally activated gene expression may necessitate sustained expression of a target gene. Unlike gene editing strategies which only require acute expression for permanent genome modification, robust CRISPR mediated transcriptional activation will likely necessitate the sustained expression of CRISPRa effector proteins to maintain stable activation. To enhance their manuscript, the authors should provide data demonstrating how long after thermal treatment gene activation is sustained. Furthermore, some endogenous genes may require days or weeks to activate if chromatin remodeling is required. The authors should clarify why they decided to perform most of their experiments at 24hrs and provide a longer time-course of viability, target activation and subsequent downregulation for a subset of endogenous genes. The authors could consider exploring the effect of pulsing thermal/ultrasound treatment for longer term activation.

Thank you for these suggestions. We have conducted new kinetics experiments in HEK293T cells to look at the time-dependent activation and cessation a GFP reporter gene. We used HEK reporter cells stably expressing TRE3G-GFP and hU6-crTet-HSP-hyperdCas12a-miniVPR-CMV-mCherry. We heated the cells and measured mean GFP fluorescence intensity and percentage of GFP positive cells at different time points after heat activation until the cells reached confluency in the culturing plate. We began to see GFP activation in just 8 hours after heat induction. Activation reached maximum around 24 hours after the initial treatment and started to decline to close to baseline around 80 hours later. As our strategy using hyperdCas12a-miniVPR works on transcriptional control of mRNA, it is temporary manipulation of gene expression without any permanent change to the genome. The temperature can also be precisely controlled to induce different extent of activation, making this strategy modular and tunable.

We have added the following discussion in the main text line 111-116:

“We conducted kinetics experiments in HEK293T TRE3G-GFP reporter cells to monitor the time-dependent activation and cessation using the HSP-hyperdCas12a-miniVPR system. We heated

the cells at 43°C for 15 min and characterized GFP activation at different time points until the cells reached confluency. We observed rapid GFP activation in 8 hours after heat induction. Gene activation reached maximum after 24 to 30 hours and started to decline to close to baseline at 80 hours after the treatment (**Extended Data Fig. 2a**).”

Extended Data Fig. 2: Characterization of time-course heat-induced hyperdCas12a-mediated gene activation in HEK293T cells. a, Left: mean GFP fluorescence intensity measured by flow cytometry at different time points after cells were treated at 43°C for 15 min. Right: percentage of GFP positive cells measured at different time points.

While the engineering of a stable system may be required for sustained CRISPRa, conversely, one challenge with stably engineered inducible systems is the issue of unintended gene activation or leakiness. This could additionally be problematic in the context of permanent genome modification (such as with the base-editor) where leaky editing could result in the accumulation of edits over time. The authors should consider exploring how tight the regulation of their base-editor is over time with sustained culture. They could consider benchmarking this against established inducible systems like tet or cumate.

Following this suggestion, we performed new experiments. In the HEK293T reporter line stably expressing TRE3G-GFP and hU6-crTet-HSP-hyperdCas12a-miniVPR-CMV-mCherry, if there is hyperdCas12a-miniVPR expression, there will be GFP expression. We cultured and maintained the cells (i.e. passaging when it was reaching confluency) for approximately 3 weeks and measured GFP fluorescence, which is the basal signal in the absence of any thermal treatment. We found that throughout the culturing period, there was little change in the mean fluorescence intensity or percentage of GFP positive cells, suggesting that there was no hyperdCas12a-miniVPR expression or very little leakiness from HSP promoter.

We have added the following description in the main text line 116-120:

“To characterize basal HSP-hyperdCas12a-miniVPR expression without heat treatment, we cultured and maintained the reporter cells for 3 weeks. Over this period, the mean fluorescence intensity or percentage of GFP positive cells did not increase, suggesting that there was no leaky hyperdCas12a-miniVPR expression from the HSP promoter (**Extended Data Fig. 2b**)”

Extended Figure 2

b

Extended Data Fig. 2 Characterization of time-course heat-induced hyperdCas12a-mediated gene activation in HEK293T cells. b, Left: mean GFP fluorescence intensity in HEK cells expressing hU6-crTet-HSP-hyperdCas12a-miniVPR-CMV-mCherry without any thermal treatment, measured over a course of 3 week. Right: percentage of GFP positive cells.

For the TRE3G-fluorescent reporter the authors should clarify the number of predicted crRNA binding sites in the text and consider including a schematic in the supplement. Assuming the authors used the full length 7X binding promoter, do the authors believe that this explains the relative efficacy of the reporter activation vs endogenous gene activation? The authors attempted to activate multiple genes with a dual gRNA tandem array but have the authors explored multiple gRNAs for the same gene as a strategy to enhance activation?

The reporter cell line TRE3G-GFP or mCherry contains 7 TetR binding sites in the promoter region, which can explain the better efficacy of heat-induced activation compared to endogenous genes. For IL2, we first tested with one guide and the activation level was low. When we added a second guide, the activation was much higher, suggesting that the activation can be enhanced by using more than one guide or more copies of the same guide. We agree that having more guide RNAs may generally lead to better gene activation. Because of this, we believe it is an advantage to use Cas12a as we did here. Compared to Cas9, it is facile to use Cas12a by designing a guide RNA array containing multiple guides that target either a single gene or multiple genes. This can potentially improve the gene regulation activity in vitro and in vivo.

We have added the following description in the main text in line 407-411:

“We observed robust reporter gene activation in HIFU treated cells but compromised performance in endogenous gene activation. The TRE3G promoter contains 7 copies of TetR binding sites, which the guide RNA can recognize and enable reporter gene expression; whereas for guides targeting endogenous genes, there is only one single binding site, thus limiting transcriptional activation.”

Thermal exposure in cells will likely activate endogenous heat-shock response pathways. The authors should discuss how this may influence the application of this technology. For all main figures the authors should include non-targeting gRNA controls to demonstrate that target gene activation is specifically the result of gRNA targeting and not heat exposure (e.g. Fig 1d,e; Fig2b.; Fig 3).

Thanks to the reviewer for pointing out the effect on endogenous heat-shock response pathways. In our approach, we purposely chose a sensitive promoter and milder hyperthermia condition (43°C for 15 min), which have been used for thermosensitive ectopic transgene expression. We also limited the elevation in temperature at a defined point to minimize any global effect. The fact that HIFU has been regularly applied on human subjects for treatments like tumor ablation suggests

that it is safe and does not have any long-term effect overall. In the case of tumor killing, heat shock response is generally considered immunostimulatory and thus reduces tumor growth. Nonetheless, it is crucial to find the optimal thermal/HIFU parameters to strike a balance between cell viability and activation effect. Following the suggestion, we have used a negative control guide RNA (crLacZ) as non-targeting guide and compared gene/protein expression at 37°C and elevated temperatures (42°C). We found that there was no expression in the target genes. The data are included in Extended Data Figure 3, 4b, 5a, 5b, and Figure 2c, 3b, 3c, 3d.

We have added the following description in the main text line 394-404:

“We chose a sensitive promoter and mild hyperthermia conditions (43°C for 15 min), which have been used by others conducting similar experiments for thermosensitive gene activation^{15,30}. HIFU-stimulated hyperthermia (with temperature increases of a few degrees) is under study for the release of drugs from temperature-sensitive delivery vehicles and the temperature increase can be monitored and controlled in real-time⁵³⁻⁵⁵. HIFU ablation (achieving a temperature above 60°C) is approved in the United States for the treatment of bone metastases, prostate cancer, uterine fibroids and neurological diseases, and at sites around the world is approved for the treatment of breast, pancreatic and liver tumors^{38,39,56}. With thermal ablation, enhanced heat shock protein expression and antigen release are generally considered immunostimulatory and reduces tumor growth⁴⁰⁻⁴¹. For the purpose of immune stimulation, it is crucial to find optimal thermal/HIFU parameters to strike a balance between cell viability and activation effect.”

Overall, the efficacy of the ultrasound mediated gene activation seems moderate both when compared directly with thermal activation in vitro as well as induction of the CRISPRa reporter in vivo. Practically this could limit the application of this technology more broadly without further optimization. Since the use of ultrasound in this manuscript is somewhat limited the authors should remove it from their title or demonstrate further optimization of this technology. Likewise, the use of base editing is limited to targeting of a reporter in vitro. The authors should remove this from their title or expand their experiments to include the editing and analysis of endogenous loci.

Our work develops a novel strategy combining focused ultrasound and CRISPR-Cas tools for multiplexed gene activation and base editing. Focused ultrasound has been used for activating individual genes, but multiplexed gene activation is more relevant in cellular engineering and reprogramming in which many genes are involved. We have now demonstrated ultrasound-mediated activation of four genes simultaneously, which has never been done before. We hope to provide a versatile and modular tool for scientists to screen for a combination of genes using guide array at defined location and time using ultrasound. As in vivo CRISPR screening is becoming popular, we believe that our approach to introduce ultrasound as a precise control would allow for better understanding of cell and disease biology in a context-specific manner.

We observed less gene activation or base editing effect with ultrasound and hypothesized that this can be improved by designing better guides or using more copies of the same guide or targeting any genes that are upstream of signaling cascade. In fact, depending on different contexts, the fold of endogenous gene expression might not need to be extremely high to have a biological or physiological effect. For example, simultaneous activation of multiple genes might exhibit a synergistic effect compared to activating a single gene, which is a topic of high interest for future cell engineering studies.

Viability following thermal exposure is clearly a challenge but it is unclear how cell viability is impacted by ultrasound treatment. The authors should clearly evaluate this in their in vitro model.

In our *in vitro* ultrasound experiment, we exposed the cells to 43°C for 15 min by either incubation in a thermocycler or focused ultrasound and compared the viability to cells that were incubated at 37°C in an incubator. We found that viability of HIFU- and thermocycler-treated cells was around 80% and 60% respectively, compared to the non-treated cells which was normalized to 100%.

We have added some description in the main text line 295-301:

“A side-by-side comparison of HIFU and treatment in the thermal cyler indicated that HIFU-treated gene activation and base editing were less efficient (**Fig. 2b, Fig. 4b-d, Extended Data Fig. 6c-d**). Although the samples were raised to the same temperature with HIFU- or thermocycler-based treatment, there was a slight decrease in cell viability with HIFU treatment, likely due to the mechanical effects of acoustic waves on isolated cells, which possibly explained the weaker activation (**Extended Data Fig. 6c**).”

Extended Data Fig. 6. HSP-hyperdCas12a is activated by HIFU *in vitro*. c, Right: Normalized cell viability after thermal treatment by HIFU or thermocycler at 43°C for 15 min. All measured values were normalized to the average percentage of live cells of samples treated at 37°C.

Specific points:

To increase the accessibility and clarity of this work please provide sequence information for critical vectors as .gb or annotated word files.

We have now added a section of sequence information in supplementary information (**Supplemental Table 1 and 2**).

- In particular, the primers listed in the methods section for heat shock promoter cloning could be predicted to amplify both the HSPA6 promoter and upstream of the similar HSPA7 pseudogene. A full vector sequence would clarify which sequence was used and enable these studies to be reproduced more easily.

We have added the sequence information in supplementary information (**Supplemental Table 1**). We will deposit all relevant constructs too Addgene.

- As mentioned previously, the sequence of the TRE3G-fluorescent reporter would be useful to clarify how many predicted crRNA target sites are retained in the promoter.

We have added the sequencing information in supplementary information (**Supplemental Table 1**) and described in Methods.

Appropriate statistical tests should be added throughout and indicated within the figure and/or figure legends.

We have added a table of statistical values in supplementary information (**Supplemental Table 3**) and figure legends.

Where possible for flow cytometry data it would be helpful to include both % positive and MFI information.

We have included both MFI and percentage of positive cells for fluorescent reporter experiments in main figures or extended figures (**Figures 1b, 2c, 3b-c, 4b, 4d, Extended Data Figure 1b-d, 2a-b, 4b-c, 5a-b, 6c-d**).

Figure 1b very usefully specifies the method of expression (viral vs stable integration) in the schematic. The authors should do this throughout the manuscript.

We have indicated the method of expression for all the constructs used in **Figures 1c, 2a-c, 3c and Extended Data Fig. 1a**.

In the description of the multiplex gRNA experiments (lines 142/143) the text reads “one guide for INFG and 2 guides for IL2”. This appears to be inconsistent with the schematic in Fig 1e which shows 1 gRNA for each gene (2 gRNAs total). The authors should clarify their approach.

The guide array contains 1 guide for IFNg and 2 guides of IL2. The schematic is a simplified demonstration for targeting 2 genes not 2 guides. We have changed the schematic in **Figure 1c** with one rhombus representing one guide.

For similar assays the authors should be consistent in the labeling of their graphs as Mean fluorescence or protein expression A.U.

We have now standardized our labeling throughout the manuscript.

For clarity the authors should make clear in the text and/or figure legends how experimental data is normalized.

We have added descriptions of how data are normalized in the relevant figure legend (**Figure 1b and Extended Data Figure 1b, 5b, 6c**).

For supplemental 1b the references to the left/right panels in the legend seem to be reversed.

We have changed the figure legend in **Extended Data Figure 1b** to

“b, Right: mean GFP fluorescence intensity measured by flow cytometry when transduced HEK cells were treated at 42°C or 43°C for 15, 30 or 45 min. Left: percentage of live cells normalized to the unheated control (i.e., 37°C).”

Reviewers' Comments:

Reviewer #1:

Remarks to the Author:

The authors have addressed my concerns. This is a nice study that adds to the field.

Reviewer #2:

Remarks to the Author:

The authors have adequately addressed my concerns, and I congratulate them on their revised manuscript.

Reviewer #3:

Remarks to the Author:

The authors have produced a revised manuscript which includes additional experiments demonstrating the simultaneous activation of 4 target genes, as well as an examination of the kinetics and duration of reporter gene activation. Conceptually, sonogenetic regulation of CRISPR activity is intriguing and could represent an interesting paradigm for non-invasive, targeted gene regulation. While the authors' revisions have strengthened the manuscript, some additional changes should be considered. See attached document.

Overall summary of the revision

A summary of major revisions is as follows:

- We have characterized the ability of our system to simultaneously activate multiple endogenous genes under the control of heat treatment (**Fig. 2b**) or ultrasound (**Fig. 4c**). We demonstrated that the system could efficiently control up to four endogenous genes, a capability not demonstrated previously.
- We carried out detailed kinetic characterization of our system to quantify the duration of target gene activation after stimulus retrieval (**Extended Data Fig. 2a**), concluding rapid induction and temporary expression of target genes.
- We performed a long time-course experiment, over three weeks, to characterize basal HSP-induced hyperdCas12a-miniVPR expression without heat treatment (**Extended Data Fig. 2b**), concluding that our system showed almost no basal leakage expression of the effector molecule.
- We've revised the main text to add new citations and address reviewers' questions.

Reviewer #3 (Remarks to the Author):

The authors have produced a revised manuscript which includes additional experiments demonstrating the simultaneous activation of 4 target genes, as well as an examination of the kinetics and duration of reporter gene activation. Conceptually, sonogenetic regulation of CRISPR activity is intriguing and could represent an interesting paradigm for non-invasive, targeted gene regulation. While the authors' revisions have strengthened the manuscript, some additional changes should be considered.

Ultrasound-mediated sonogenetic control of genome regulation and base editing

In this manuscript the authors present a novel strategy for the regulation of endogenous gene expression through the coupling of thermogenetic transcriptional control and CRISPR technologies. Utilizing a previously described, hyperdCas12a-miniVPR effector under the control of a heat shock promoter, the authors are able to transcriptionally activate the expression of a GFP reporter in a temperature dependent manner. The authors subsequently demonstrate the activation of several endogenous target genes in multiple cellular contexts and using dual gRNA arrays. In addition to CRISPRa, the authors also explore thermoregulation in the context of a base-editing reporter. Finally, the authors demonstrate the use of ultrasound as a strategy for thermal induction of CRISPRa in both in vitro cell culture and an in vivo murine implantation model.

The challenge of achieving robust, precise, non-invasive gene regulation is critical for manipulating complex cellular models and for the safety and efficacy of cell-based therapeutics. The authors of this manuscript demonstrate a creative approach towards these ends and this work provides a compelling proof-of-concept for future platform development. While conceptually intriguing we feel that this work, as presented, is underdeveloped for publication in Nature Communications without expanded studies.

Broad critiques:

Many of the potential research and clinical applications for thermally activated gene expression may necessitate sustained expression of a target gene. Unlike gene editing strategies which only require acute expression for permanent genome modification, robust CRISPR mediated transcriptional activation will likely necessitate the sustained expression of CRISPRa effector proteins to maintain stable activation. To enhance their manuscript, the authors should provide data demonstrating how long after thermal treatment gene activation is sustained. Furthermore, some endogenous genes may require days or weeks to activate if chromatin remodeling is required. The authors should clarify why they decided to perform most of their experiments at 24hrs and provide a longer time-course of viability, target activation and subsequent downregulation for a subset of endogenous genes. The authors could consider exploring the effect of pulsing thermal/ultrasound treatment for longer term activation.

Thank you for these suggestions. We have conducted new kinetics experiments in HEK293T cells to look at the time-dependent activation and cessation a GFP reporter gene. We used HEK reporter cells stably expressing TRE3G-GFP and hU6-crTet-HSP-hyperdCas12a-miniVPR-CMV-mCherry. We heated the cells and measured mean GFP fluorescence intensity and percentage of GFP positive cells at different time points after heat activation until the cells reached confluency in the culturing plate. We began to see GFP activation in just 8 hours after heat induction. Activation reached maximum around 24 hours after the initial treatment and started to decline to close to baseline around 80 hours later. As our strategy using hyperdCas12a-miniVPR works on transcriptional control of mRNA, it is temporary manipulation of gene expression without any permanent change to the genome. The temperature can also be precisely controlled to induce different extent of activation, making this strategy modular and tunable.

We have added the following discussion in the main text line 111-116:

“We conducted kinetics experiments in HEK293T TRE3G-GFP reporter cells to monitor the time-dependent activation and cessation using the HSP-hyperdCas12a-miniVPR system. We heated the cells at 43°C for 15 min and characterized GFP activation at different time points until the cells reached confluency. We observed rapid GFP activation in 8 hours after heat induction. Gene activation reached maximum after 24 to 30 hours and started to decline to close to baseline at 80 hours after the treatment (**Extended Data Fig. 2a**).”

Extended Data Fig. 2: Characterization of time-course heat-induced hyperdCas12a-mediated gene activation in HEK293T cells. a, Left: mean GFP fluorescence intensity measured by flow cytometry at different time points after cells were treated at 43°C for 15 min. Right: percentage of GFP positive cells measured at different time points.

This experiment is very useful for understanding the kinetics and stability of reporter gene induction. The authors should include a statement in the text or discussion to discuss if they believe the rapid downregulation of target gene expression is reflective of what would be expected in the context of an endogenous target gene and the broader implications/limitations that may represent for experimental and therapeutic applications.

While the engineering of a stable system may be required for sustained CRISPRa, conversely, one challenge with stably engineered inducible systems is the issue of unintended gene activation or leakiness. This could additionally be problematic in the context of permanent genome modification (such as with the base-editor) where leaky editing could result in the accumulation of edits over time. The authors should consider exploring how tight the regulation of their base-editor is over time with sustained culture. They could consider benchmarking this against established inducible systems like tet or cumate.

Following this suggestion, we performed new experiments. In the HEK293T reporter line stably expressing TRE3G-GFP and hU6-crTet-HSP-hyperdCas12a-miniVPR-CMV-mCherry, if there is hyperdCas12a-miniVPR expression, there will be GFP expression. We cultured and maintained the cells (i.e. passaging when it was reaching confluency) for approximately 3 weeks and measured GFP fluorescence, which is the basal signal in the absence of any thermal treatment. We found that throughout the culturing period, there was little change in the mean fluorescence intensity or percentage of GFP positive cells, suggesting that there was no hyperdCas12a-miniVPR expression or very little leakiness from HSP promoter.

We have added the following description in the main text line 116-120:

“To characterize basal HSP-hyperdCas12a-miniVPR expression without heat treatment, we cultured and maintained the reporter cells for 3 weeks. Over this period, the mean fluorescence intensity or percentage of GFP positive cells did not increase, suggesting that there was no leaky hyperdCas12a-miniVPR expression from the HSP promoter (**Extended Data Fig. 2b**)”

Extended Figure 2

Extended Data Fig. 2 Characterization of time-course heat-induced hyperdCas12a-mediated gene activation in HEK293T cells. b, Left: mean GFP fluorescence intensity in HEK cells expressing hU6-crTet-HSP-hyperdCas12a-miniVPR-CMV-mCherry without any thermal treatment, measured over a course of 3 week. Right: percentage of GFP positive cells.

These experiments add to this manuscript and suggest that leaky expression likely not a problem with respect to long-term use of the stable CRISPRa system. As highlighted previously, due to the permanent nature of base editing, the accumulation of unintended ‘leaky’ genomic edits over time would likely present a larger technical issue for this technology. If the authors cannot provide data

to address leakiness in the context of the base editor, they should discuss in the text how leakiness may be more problematic in that context.

For the TRE3G-fluorescent reporter the authors should clarify the number of predicted crRNA binding sites in the text and consider including a schematic in the supplement. Assuming the authors used the full length 7X binding promoter, do the authors believe that this explains the relative efficacy of the reporter activation vs endogenous gene activation? The authors attempted to activate multiple genes with a dual gRNA tandem array but have the authors explored multiple gRNAs for the same gene as a strategy to enhance activation?

The reporter cell line TRE3G-GFP or mCherry contains 7 TetR binding sites in the promoter region, which can explain the better efficacy of heat-induced activation compared to endogenous genes. For IL2, we first tested with one guide and the activation level was low. When we added a second guide, the activation was much higher, suggesting that the activation can be enhanced by using more than one guide or more copies of the same guide. We agree that having more guide RNAs may generally lead to better gene activation. Because of this, we believe it is an advantage to use Cas12a as we did here. Compared to Cas9, it is facile to use Cas12a by designing a guide RNA array containing multiple guides that target either a single gene or multiple genes. This can potentially improve the gene regulation activity in vitro and in vivo.

We have added the following description in the main text in line 407-411:

“We observed robust reporter gene activation in HIFU treated cells but compromised performance in endogenous gene activation. The TRE3G promoter contains 7 copies of TetR binding sites, which the guide RNA can recognize and enable reporter gene expression; whereas for guides targeting endogenous genes, there is only one single binding site, thus limiting transcriptional activation.”

These modifications to the manuscript address the concerns raised.

Thermal exposure in cells will likely activate endogenous heat-shock response pathways. The authors should discuss how this may influence the application of this technology. For all main figures the authors should include non-targeting gRNA controls to demonstrate that target gene activation is specifically the result of gRNA targeting and not heat exposure (e.g. Fig 1d,e; Fig2b.; Fig 3).

Thanks to the reviewer for pointing out the effect on endogenous heat-shock response pathways. In our approach, we purposely chose a sensitive promoter and milder hyperthermia condition (43°C for 15 min), which have been used for thermosensitive ectopic transgene expression. We also limited the elevation in temperature at a defined point to minimize any global effect. The fact that HIFU has been regularly applied on human subjects for treatments like tumor ablation suggests that it is safe and does not have any long-term effect overall. In the case of tumor killing, heat shock response is generally considered immunostimulatory and thus reduces tumor growth. Nonetheless, it is crucial to find the optimal thermal/HIFU parameters to strike a balance between cell viability and activation effect. Following the suggestion, we have used a negative control guide RNA (crLacZ) as non-targeting guide and compared gene/protein expression at 37°C and elevated temperatures (42°C). We found that there was no expression in the target genes. The data are included in Extended Data Figure 3, 4b, 5a, 5b, and Figure 2c, 3b, 3c, 3d.

We have added the following description in the main text line 394-404:

“We chose a sensitive promoter and mild hyperthermia conditions (43°C for 15 min), which have been used by others conducting similar experiments for thermosensitive gene activation^{15,30}. HIFU-stimulated hyperthermia (with temperature increases of a few degrees) is under study for the release of drugs from temperature-sensitive delivery vehicles and the temperature increase can be monitored and controlled in real-time⁵³⁻⁵⁵. HIFU ablation (achieving a temperature above 60°C) is approved in the United States for the treatment of bone metastases, prostate cancer, uterine fibroids and neurological diseases, and at sites around the world is approved for the treatment of breast, pancreatic and liver tumors^{38,39,56}. With thermal ablation, enhanced heat shock protein expression and antigen release are generally considered immunostimulatory and reduces tumor growth⁴⁰⁻⁴¹. For the purpose of immune stimulation, it is crucial to find optimal thermal/HIFU parameters to strike a balance between cell viability and activation effect.”

The addition of these controls and text strengthens the manuscript. Extended data 3 would be further strengthened with the addition of IL18 as that target gene is used throughout the paper.

Overall, the efficacy of the ultrasound mediated gene activation seems moderate both when compared directly with thermal activation in vitro as well as induction of the CRISPRa reporter in vivo. Practically this could limit the application of this technology more broadly without further optimization. Since the use of ultrasound in this manuscript is somewhat limited the authors should remove it from their title or demonstrate further optimization of this technology. Likewise, the use of base editing is limited to targeting of a reporter in vitro. The authors should remove this from their title or expand their experiments to include the editing and analysis of endogenous loci.

Our work develops a novel strategy combining focused ultrasound and CRISPR-Cas tools for multiplexed gene activation and base editing. Focused ultrasound has been used for activating individual genes, but multiplexed gene activation is more relevant in cellular engineering and reprogramming in which many genes are involved. We have now demonstrated ultrasound-mediated activation of four genes simultaneously, which has never been done before. We hope to provide a versatile and modular tool for scientists to screen for a combination of genes using guide array at defined location and time using ultrasound. As in vivo CRISPR screening is becoming popular, we believe that our approach to introduce ultrasound as a precise control would allow for better understanding of cell and disease biology in a context-specific manner.

We observed less gene activation or base editing effect with ultrasound and hypothesized that this can be improved by designing better guides or using more copies of the same guide or targeting any genes that are upstream of signaling cascade. In fact, depending on different contexts, the fold of endogenous gene expression might not need to be extremely high to have a biological or physiological effect. For example, simultaneous activation of multiple genes might exhibit a synergistic effect compared to activating a single gene, which is a topic of high interest for future cell engineering studies.

Original title: Ultrasound-mediated sonogenetic control of genome regulation and base editing

Current title: Sonogenetic control of multiplexed genome regulation and base editing

The removal of “ultrasound-mediated” and the addition of “multiplexed” is representative of the data shared in the paper. I will leave it to the editors to decide if the limited use of base editing warrants its inclusion in the title.

Viability following thermal exposure is clearly a challenge but it is unclear how cell viability is impacted by ultrasound treatment. The authors should clearly evaluate this in their *in vitro* model. In our *in vitro* ultrasound experiment, we exposed the cells to 43°C for 15 min by either incubation in a thermocycler or focused ultrasound and compared the viability to cells that were incubated at 37°C in an incubator. We found that viability of HIFU- and thermocycler-treated cells was around 80% and 60% respectively, compared to the non-treated cells which was normalized to 100%.

We have added some description in the main text line 295-301:

“A side-by-side comparison of HIFU and treatment in the thermal cycler indicated that HIFU-treated gene activation and base editing were less efficient (**Fig. 2b, Fig. 4b-d, Extended Data Fig. 6c-d**). Although the samples were raised to the same temperature with HIFU- or thermocycler-based treatment, there was a slight decrease in cell viability with HIFU treatment, likely due to the mechanical effects of acoustic waves on isolated cells, which possibly explained the weaker activation (**Extended Data Fig. 6c**).”

Extended Data Fig. 6. HSP-hyperdCas12a is activated by HIFU *in vitro*. c, Right: Normalized cell viability after thermal treatment by HIFU or thermocycler at 43°C for 15 min. All measured values were normalized to the average percentage of live cells of samples treated at 37°C.

Specific points:

To increase the accessibility and clarity of this work please provide sequence information for critical vectors as .gb or annotated word files.

We have now added a section of sequence information in supplementary information (**Supplemental Table 1 and 2**).

The concern has been addressed.

• In particular, the primers listed in the methods section for heat shock promoter cloning could be predicted to amplify both the HSPA6 promoter and upstream of the similar HSPA7 pseudogene. A full vector sequence would clarify which sequence was used and enable these studies to be reproduced more easily.

We have added the sequence information in supplementary information (**Supplemental Table 1**). We will deposit all relevant constructs too Addgene.

The concern has been addressed.

• As mentioned previously, the sequence of the TRE3G-fluorescent reporter would be useful to clarify how many predicted crRNA target sites are retained in the promoter.

We have added the sequencing information in supplementary information (**Supplemental Table 1**) and described in Methods.

The concern has been addressed.

Appropriate statistical tests should be added throughout and indicated within the figure and/or figure legends.

We have added a table of statistical values in supplementary information (**Supplemental Table 3**) and figure legends.

The editor should decide if this is consistent with journal policy.

Where possible for flow cytometry data it would be helpful to include both % positive and MFI information.

We have included both MFI and percentage of positive cells for fluorescent reporter experiments in main figures or extended figures (**Figures 1b, 2c, 3b-c, 4b, 4d, Extended Data Figure 1b-d, 2a-b, 4b-c, 5a-b, 6c-d**).

The changes mentioned above by the authors are not reflected in the current manuscript.

Figure 1b includes MFI but not % positive

Figure 2c only includes %GFP,

Figure 3b only includes % GFP

Figure 3c only includes MFI

Figure 4b only includes % GFP

Figure 4d only includes % GFP

Extended data 4b only includes MFI- As this figure only includes data from HEK cells Jurkats should not be referenced in the legend.

Extended data 5a only includes MFI

Extended data 5b only includes % GFP

Extended data 6c includes only MFI

If both measures cannot be provided then the authors should be consistent throughout the manuscript.

Figure 1b very usefully specifies the method of expression (viral vs stable integration) in the schematic. The authors should do this throughout the manuscript.

We have indicated the method of expression for all the constructs used in **Figures 1c, 2a-c, 3c and Extended Data Fig. 1a**.

The concern has been addressed.

In the description of the multiplex gRNA experiments (lines 142/143) the text reads “one guide for INFG and 2 guides for IL2”. This appears to be inconsistent with the schematic in Fig 1e which shows 1 gRNA for each gene (2 gRNAs total). The authors should clarify their approach.

The guide array contains 1 guide for IFN γ and 2 guides of IL2. The schematic is a simplified demonstration for targeting 2 genes not 2 guides. We have changed the schematic in **Figure 1c** with one rhombus representing one guide.

The concern has been addressed.

For similar assays the authors should be consistent in the labeling of their graphs as Mean fluorescence or protein expression A.U.

We have now standardized our labeling throughout the manuscript.

Throughout the paper, figures and legend the use of NT/crLacZ and INFG/INF γ is inconsistent.

For clarity the authors should make clear in the text and/or figure legends how experimental data is normalized.

We have added descriptions of how data are normalized in the relevant figure legend (**Figure 1b and Extended Data Figure 1b, 5b, 6c**).

These additions further clarify the authors methods. For **figure 2c** and like experiments, however, it is still unclear if % GFP+ represents %GFP/all cells or % GFP/ mCherry+ cells.

For supplemental 1b the references to the left/right panels in the legend seem to be reversed.

We have changed the figure legend in **Extended Data Figure 1b** to

“b, Right: mean GFP fluorescence intensity measured by flow cytometry when transduced HEK cells were treated at 42°C or 43°C for 15, 30 or 45 min. Left: percentage of live cells normalized to the unheated control (i.e., 37°C).”

The concern has been addressed.

Overall summary of the revision

A summary of major revisions (the first round) is as follows:

- We have characterized the ability of our system to simultaneously activate multiple endogenous genes under the control of heat treatment (**Fig. 2b**) or ultrasound (**Fig. 4c**). We demonstrated that the system could efficiently control up to four endogenous genes, a capability not demonstrated previously. • We carried out detailed kinetic characterization of our system to quantify the duration of target gene activation after stimulus retrieval (**Extended Data Fig. 2a**), concluding rapid induction and temporary expression of target genes.
- We performed a long time-course experiment, over three weeks, to characterize basal HSP-induced hyperdCas12a-miniVPR expression without heat treatment (**Extended Data Fig. 2b**), concluding that our system showed almost no basal leakage expression of the effector molecule.
- We've revised the main text to add new citations and address reviewers' questions.

In the second round of revision, we've addressed all remaining comments from Reviewer 3. A summary of revisions (the second round) is as follows:

- We've added new data for IL18 and IL7 (**Extended Data Fig. 3**).
- We've added discussion of the ultrasound-mediated gene expression kinetics for endogenous genes in the Discussion.
- We've added text to describe the potential impacts on long-term base editing due to leaky basal expression of the HSP-Cas system.
- We've included data with both MFI values and GFP+% characterization (in main figures or Extended Data Figures)
- We've added description of how GFP+% cells were calculated in the Figure legends.
- In this second response, we have added to the previous response in purple.

Reviewer #3 (Remarks to the Author):

The authors have produced a revised manuscript which includes additional experiments demonstrating the simultaneous activation of 4 target genes, as well as an examination of the kinetics and duration of reporter gene activation. Conceptually, sonogenetic regulation of CRISPR activity is intriguing and could represent an interesting paradigm for non-invasive, targeted gene regulation. While the authors' revisions have strengthened the manuscript, some additional changes should be considered.

Response: We thank the reviewer for reading the revised manuscript carefully and recognizing the value of the study. We have addressed all remaining questions and provide a point-by-point response here.

Ultrasound-mediated sonogenetic control of genome regulation and base editing

In this manuscript the authors present a novel strategy for the regulation of endogenous gene expression through the coupling of thermogenetic transcriptional control and CRISPR technologies. Utilizing a previously described, hyperdCas12a-miniVPR effector under the control of a heat shock promoter, the authors are able to transcriptionally activate the expression of a GFP reporter in a temperature dependent manner. The authors subsequently demonstrate the activation of several endogenous target genes in multiple cellular contexts and using dual gRNA arrays. In addition to CRISPRa, the authors also explore thermoregulation in the context of a base-editing reporter. Finally, the authors demonstrate the use of ultrasound as a strategy for thermal induction of CRISPRa in both in vitro cell culture and an in vivo murine implantation model.

The challenge of achieving robust, precise, non-invasive gene regulation is critical for manipulating complex cellular models and for the safety and efficacy of cell-based therapeutics. The authors of this manuscript demonstrate a creative approach towards these ends and this work provides a compelling proof-of-concept for future platform development. While conceptually intriguing we feel that this work, as presented, is underdeveloped for publication in Nature Communications without expanded studies.

Broad critiques:

Many of the potential research and clinical applications for thermally activated gene expression may necessitate sustained expression of a target gene. Unlike gene editing strategies which only require acute expression for permanent genome modification, robust CRISPR mediated transcriptional activation will likely necessitate the sustained expression of CRISPRa effector proteins to maintain stable activation. To enhance their manuscript, the authors should provide data demonstrating how long after thermal treatment gene activation is sustained. Furthermore, some endogenous genes may require days or weeks to activate if chromatin remodeling is required. The authors should clarify why they decided to perform most of their experiments at 24hrs and provide a longer time-course of viability, target activation and subsequent downregulation for a subset of endogenous genes. The authors could consider exploring the effect of pulsing thermal/ultrasound treatment for longer term activation.

Thank you for these suggestions. We have conducted new kinetics experiments in HEK293T cells to look at the time-dependent activation and cessation a GFP reporter gene. We used HEK reporter cells stably expressing TRE3G-GFP and hU6-crTet-HSP-hyperdCas12a-miniVPR-CMVmCherry. We heated the cells and measured mean GFP fluorescence intensity and percentage of GFP positive cells at different time points after heat activation until the cells reached confluency in the culturing plate. We began to see GFP activation in just 8 hours after heat induction. Activation reached maximum around 24 hours after the initial treatment and started to decline to close to baseline around 80 hours later. As our strategy using hyperdCas12a-miniVPR works on transcriptional control of mRNA, it is temporary manipulation of gene expression without any permanent change to the genome. The temperature can also be precisely controlled to induce different extent of activation, making this strategy modular and tunable.

We have added the following discussion in the main text line 111-116:

“We conducted kinetics experiments in HEK293T TRE3G-GFP reporter cells to monitor the time-dependent activation and cessation using the HSP-hyperdCas12a-miniVPR system. We heated the cells at 43°C for 15 min and characterized GFP activation at different time points until the cells reached confluency. We observed rapid GFP activation in 8 hours after heat induction. Gene activation reached maximum after 24 to 30 hours and started to decline to close to baseline at 80 hours after the treatment (**Extended Data Fig. 2a**).”

Extended Data Fig. 2: Characterization of time-course heat-induced hyperdCas12a-mediated gene activation in HEK293T cells. a, Left: mean GFP fluorescence intensity measured by flow cytometry at different time points after cells were treated at 43°C for 15 min. Right: percentage of GFP positive cells measured at different time points.

This experiment is very useful for understanding the kinetics and stability of reporter gene induction. The authors should include a statement in the text or discussion to discuss if they believe the rapid downregulation of target gene expression is reflective of what would be expected in the context of an endogenous target gene and the broader implications/limitations that may represent for experimental and therapeutic applications.

Response: We have provided a statement regarding our expectation on endogenous target gene kinetics using ultrasound control and its broader implications. The text is included (Line 394-401) and included here:

“In the current study, we detected a rapid downregulation of target gene expression (**Extended Data Fig. 2**), implying that our ultrasound-induced transcriptional activation system induces short-term and reversible changes in gene expression. It is worth noting that the kinetics of activation and cessation of an endogenous gene upon stimulation can vary significantly depending on the cell type and the specific gene being targeted. The ultrasound stimulus offers the advantage of repeatable administration in a non-invasive manner, with precise control over its duration and amplitude. This suggests the potential for enhancing the initial upregulation of the target gene through subsequent rounds of ultrasound stimulation.”

While the engineering of a stable system may be required for sustained CRISPRa, conversely, one challenge with stably engineered inducible systems is the issue of unintended gene activation or leakiness. This could additionally be problematic in the context of permanent genome modification (such as with the base-editor) where leaky editing could result in the accumulation of edits over time. The authors should consider exploring how tight the regulation of their baseeditor

is over time with sustained culture. They could consider benchmarking this against established inducible systems like tet or cumate.

Following this suggestion, we performed new experiments. In the HEK293T reporter line stably expressing TRE3G-GFP and hU6-crTet-HSP-hyperdCas12a-miniVPR-CMV-mCherry, if there is hyperdCas12a-miniVPR expression, there will be GFP expression. We cultured and maintained the cells (i.e. passaging when it was reaching confluency) for approximately 3 weeks and measured GFP fluorescence, which is the basal signal in the absence of any thermal treatment. We found that throughout the culturing period, there was little change in the mean fluorescence intensity or percentage of GFP positive cells, suggesting that there was no hyperdCas12a-miniVPR expression or very little leakiness from HSP promoter.

We have added the following description in the main text line 116-120:

“To characterize basal HSP-hyperdCas12a-miniVPR expression without heat treatment, we cultured and maintained the reporter cells for 3 weeks. Over this period, the mean fluorescence intensity or percentage of GFP positive cells did not increase, suggesting that there was no leaky hyperdCas12a-miniVPR expression from the HSP promoter (**Extended Data Fig. 2b**)”

Extended Figure 2

Extended Data Fig. 2 Characterization of time-course heat-induced hyperdCas12a-mediated gene activation in HEK293T cells. b, Left: mean GFP fluorescence intensity in HEK cells expressing hU6-crTet-HSP-hyperdCas12a-miniVPR-CMV-mCherry without any thermal treatment, measured over a course of 3 week. Right: percentage of GFP positive cells.

These experiments add to this manuscript and suggest that leaky expression likely not a problem with respect to long-term use of the stable CRISPRa system. As highlighted previously, due to the permanent nature of base editing, the accumulation of unintended ‘leaky’ genomic edits over time would likely present a larger technical issue for this technology. If the authors cannot provide data to address leakiness in the context of the base editor, they should discuss in the text how leakiness may be more problematic in that context.

Response: We have provided a statement regarding the leakiness in the context of the base editor. The text is included (Line 186-191) and included here:

“While our observations over a three-week period did not reveal leaky HSP-Cas expression or target gene activation in the absence of heat treatment, it is possible that long-term accumulation of low amplitude leakiness of the base editing system may lead to more pronounced, permanent changes on the genome. However, this can be mitigated by titrating the expression level of HSP-Cas in transduced cells to minimize its background expression at physiological temperatures.”

For the TRE3G-fluorescent reporter the authors should clarify the number of predicted crRNA binding sites in the text and consider including a schematic in the supplement. Assuming the authors used the full length 7X binding promoter, do the authors believe that this explains the relative efficacy of the reporter activation vs endogenous gene activation? The authors attempted to activate multiple genes with a dual gRNA tandem array but have the authors explored multiple gRNAs for the same gene as a strategy to enhance activation?

The reporter cell line TRE3G-GFP or mCherry contains 7 TetR binding sites in the promoter region, which can explain the better efficacy of heat-induced activation compared to endogenous genes. For IL2, we first tested with one guide and the activation level was low. When we added a second guide, the activation was much higher, suggesting that the activation can be enhanced by using more than one guide or more copies of the same guide. We agree that having more guide RNAs may generally lead to better gene activation. Because of this, we believe it is an advantage to use Cas12a as we did here. Compared to Cas9, it is facile to use Cas12a by designing a guide RNA array containing multiple guides that target either a single gene or multiple genes. This can potentially improve the gene regulation activity in vitro and in vivo.

We have added the following description in the main text in line 407-411:

“We observed robust reporter gene activation in HIFU treated cells but compromised performance in endogenous gene activation. The TRE3G promoter contains 7 copies of TetR binding sites, which the guide RNA can recognize and enable reporter gene expression; whereas for guides targeting endogenous genes, there is only one single binding site, thus limiting transcriptional activation.”

These modifications to the manuscript address the concerns raised.

Thermal exposure in cells will likely activate endogenous heat-shock response pathways. The authors should discuss how this may influence the application of this technology. For all main figures the authors should include non-targeting gRNA controls to demonstrate that target gene activation is specifically the result of gRNA targeting and not heat exposure (e.g. Fig 1d,e; Fig2b.; Fig 3).

Thanks to the reviewer for pointing out the effect on endogenous heat-shock response pathways. In our approach, we purposely chose a sensitive promoter and milder hyperthermia condition (43°C for 15 min), which have been used for thermosensitive ectopic transgene expression. We also limited the elevation in temperature at a defined point to minimize any global effect. The fact that HIFU has been regularly applied on human subjects for treatments like tumor ablation suggests that it is safe and does not have any long-term effect overall. In the case of tumor killing, heat shock response is generally considered immunostimulatory and thus reduces tumor growth. Nonetheless, it is crucial to find the optimal thermal/HIFU parameters to strike a balance between cell viability and activation effect. Following the suggestion, we have used a negative control guide RNA (crLacZ) as non-targeting guide and compared gene/protein expression at 37°C and elevated temperatures (42°C). We found that there was no expression in the target genes. The data are included in Extended Data Figure 3, 4b, 5a, 5b, and Figure 2c, 3b, 3c, 3d.

We have added the following description in the main text line 394-404:

“We chose a sensitive promoter and mild hyperthermia conditions (43°C for 15 min), which have been used by others conducting similar experiments for thermosensitive gene activation^{15,30}. HIFU-stimulated hyperthermia (with temperature increases of a few degrees) is under study for the release of drugs from temperature-sensitive delivery vehicles and the temperature increase can be monitored and controlled in real-time⁵³⁻⁵⁵. HIFU ablation (achieving a temperature above 60°C) is approved in the United States for the treatment of bone metastases, prostate cancer, uterine fibroids and neurological diseases, and at sites around the world is approved for the treatment of breast, pancreatic and liver tumors^{38,39,56}. With thermal ablation, enhanced heat shock protein expression and antigen release are generally considered immunostimulatory and reduces tumor growth⁴⁰⁻⁴¹. For the purpose of immune stimulation, it is crucial to find optimal thermal/HIFU parameters to strike a balance between cell viability and activation effect.”

The addition of these controls and text strengthens the manuscript. Extended data 3 would be further strengthened with the addition of IL18 as that target gene is used throughout the paper.

Response: We have now added the data quantifying the endogenous IL18 and IL7 expression in cells transfected with a non-targeting guide crLacZ or a guide that targeting IL18 or IL7 (crIL18 or crIL7), before and after heat treatment (see new **Extended Data Figure 3**).

Extended Data Fig. 3: Characterization of heat-induced hyperdCas12a-mediated gene activation for a variety of endogenous genes in HEK293T cells. Endogenous gene activation measured by ELISA (for IFN γ , IL2, CXCL10) or by immunostaining (for CD2 and CXCR4) or by qPCR (for IL18 and IL7) after cells were transfected with guide RNAs and treated at 42°C for 30 min. crLacZ: non-targeting guide. All data are shown for 1-3 independent replicates. Unpaired two-sided t test was used for statistical analysis and p-values are presented in **Supplementary Table 3**.

Overall, the efficacy of the ultrasound mediated gene activation seems moderate both when compared directly with thermal activation in vitro as well as induction of the CRISPRa reporter in vivo. Practically this could limit the application of this technology more broadly without further optimization. Since the use of ultrasound in this manuscript is somewhat limited the authors should remove it from their title or demonstrate further optimization of this technology. Likewise, the use of base editing is limited to targeting of a reporter in vitro. The authors should remove this from their title or expand their experiments to include the editing and analysis of endogenous loci.

Our work develops a novel strategy combining focused ultrasound and CRISPR-Cas tools for multiplexed gene activation and base editing. Focused ultrasound has been used for activating individual genes, but multiplexed gene activation is more relevant in cellular engineering and reprogramming in which many genes are involved. We have now demonstrated ultrasound-mediated activation of four genes simultaneously, which has never been done before. We hope to provide a versatile and modular tool for scientists to screen for a combination of genes using guide array at defined location and time using ultrasound. As in vivo CRISPR screening is becoming popular, we believe that our approach to introduce ultrasound as a precise control would allow for better understanding of cell and disease biology in a context-specific manner.

We observed less gene activation or base editing effect with ultrasound and hypothesized that this can be improved by designing better guides or using more copies of the same guide or targeting any genes that are upstream of signaling cascade. In fact, depending on different contexts, the fold of endogenous gene expression might not need to be extremely high to have a biological or physiological effect. For example, simultaneous activation of multiple genes might exhibit a synergistic effect compared to activating a single gene, which is a topic of high interest for future cell engineering studies.

Original title: Ultrasound-mediated sonogenetic control of genome regulation and base editing

Current title: Sonogenetic control of multiplexed genome regulation and base editing

The removal of “ultrasound-mediated” and the addition of “multiplexed” is representative of the data shared in the paper. I will leave it to the editors to decide if the limited use of base editing warrants its inclusion in the title.

Response: We believe including ‘base editing’ in the title will make the manuscript more appealing to the broader researchers in the field of genome engineering.

Viability following thermal exposure is clearly a challenge but it is unclear how cell viability is impacted by ultrasound treatment. The authors should clearly evaluate this in their in vitro model. In our in vitro ultrasound experiment, we exposed the cells to 43°C for 15 min by either incubation in a thermocycler or focused ultrasound and compared the viability to cells that were incubated at 37°C in an incubator. We found that viability of HIFU- and thermocycler-treated cells was around 80% and 60% respectively, compared to the non-treated cells which was normalized to 100%.

We have added some description in the main text line 295-301:

“A side-by-side comparison of HIFU and treatment in the thermal cyler indicated that HIFU-treated gene activation and base editing were less efficient (Fig. 2b, Fig. 4b-d, Extended Data Fig. 6c-d). Although the samples were raised to the same temperature with HIFU- or thermocycler-based treatment, there was a slight decrease in cell viability with HIFU treatment, likely due to the mechanical effects of acoustic waves on isolated cells, which possibly explained the weaker activation (Extended Data Fig. 6c).”

Extended Data Fig. 6. HSP-hyperdCas12a is activated by HIFU *in vitro*. c, Right: Normalized cell viability after thermal treatment by HIFU or thermocycler at 43°C for 15 min. All measured values were normalized to the average percentage of live cells of samples treated at 37°C.

Specific points:

To increase the accessibility and clarity of this work please provide sequence information for critical vectors as .gb or annotated word files.

We have now added a section of sequence information in supplementary information (**Supplemental Table 1 and 2**).

The concern has been addressed.

- In particular, the primers listed in the methods section for heat shock promoter cloning could be predicted to amplify both the HSPA6 promoter and upstream of the similar HSPA7 pseudogene. A full vector sequence would clarify which sequence was used and enable these studies to be reproduced more easily.

We have added the sequence information in supplementary information (**Supplemental Table 1**). We will deposit all relevant constructs too Addgene.

The concern has been addressed.

- As mentioned previously, the sequence of the TRE3G-fluorescent reporter would be useful to clarify how many predicted crRNA target sites are retained in the promoter.

We have added the sequencing information in supplementary information (**Supplemental Table 1**) and described in Methods.

The concern has been addressed.

Appropriate statistical tests should be added throughout and indicated within the figure and/or figure legends.

We have added a table of statistical values in supplementary information (**Supplemental Table 3**) and figure legends.

The editor should decide if this is consistent with journal policy.

Where possible for flow cytometry data it would be helpful to include both % positive and MFI information.

We have included both MFI and percentage of positive cells for fluorescent reporter experiments in main figures or extended figures (**Figures 1b, 2c, 3b-c, 4b, 4d, Extended Data Figure 1b-d, 2a-b, 4b-c, 5a-b, 6c-d**).

The changes mentioned above by the authors are not reflected in the current manuscript.

Figure 1b includes MFI but not % positive

Figure 2c only includes %GFP,

Figure 3b only includes % GFP

Figure 3c only includes MFI

Figure 4b only includes % GFP

Figure 4d only includes % GFP

Extended data 4b only includes MFI- As this figure only includes data from HEK cells Jurkats should not be referenced in the legend. Extended data 5a only includes MFI

Extended data 5b only includes % GFP

Extended data 6c includes only MFI

If both measures cannot be provided then the authors should be consistent throughout the manuscript.

Response: Due to limited space in the main text, we have moved some data to the Extended Data Figures with relevant description and discussion of the data in the main text. We have mentioned the other type of quantification (MFI vs. GFP+% cells) in the main figure legends so readers can refer to them easily. We also summarize the MFI and corresponding %GFP+ figures in the following table:

MFI	Fig. 1b	Extended Data Fig. 4b	Extended Data Fig. 5a	Fig. 3c	Extended Data Fig. 6c	Extended Data Fig. 6d
%GFP positive	Extended Data Fig. 1b	Fig. 2c	Fig. 3b	Extended Data Fig. 5b	Fig. 4b	Fig. 4d

We have corrected the figure legend in Extended Data Fig. 4a to remove Jurkat cells.

Figure 1b very usefully specifies the method of expression (viral vs stable integration) in the schematic. The authors should do this throughout the manuscript.

We have indicated the method of expression for all the constructs used in **Figures 1c, 2a-c, 3c and Extended Data Fig. 1a**.

The concern has been addressed.

In the description of the multiplex gRNA experiments (lines 142/143) the text reads “one guide for INFG and 2 guides for IL2”. This appears to be inconsistent with the schematic in Fig 1e which shows 1 gRNA for each gene (2 gRNAs total). The authors should clarify their approach.

The guide array contains 1 guide for IFNg and 2 guides of IL2. The schematic is a simplified demonstration for targeting 2 genes not 2 guides. We have changed the schematic in **Figure 1c** with one rhombus representing one guide.

The concern has been addressed.

For similar assays the authors should be consistent in the labeling of their graphs as Mean fluorescence or protein expression A.U.

We have now standardized our labeling throughout the manuscript.

Throughout the paper, figures and legend the use of NT/crLacz and INFG/INF γ is inconsistent.

Response: We use crLacZ for non-targeting guide, IFNG for the gene and IFN- γ for protein. We have updated the relevant figures and legends (**Fig. 2a, 2b, 3a, Extended Data Fig. 3, 4b, 5a**)

For clarity the authors should make clear in the text and/or figure legends how experimental data is normalized.

We have added descriptions of how data are normalized in the relevant figure legend (**Figure 1b and Extended Data Figure 1b, 5b, 6c**).

These additions further clarify the authors methods. For **figure 2c** and like experiments, however, it is still unclear if % GFP+ represents %GFP/all cells or % GFP/ mCherry+ cells.

Response: %GFP+ cells is %GFP/mCherry+ cells (i.e. cells that express our protein/guide of interest). We have added descriptions of how we calculated %GFP+ or %mCherry+ cells in the relevant figure legends (**Fig. 2c, 3b, 4b, 4d, 5d, Extended Data Fig. 1b, 1d, 2a, 2b, 4c, 5b**)

For supplemental 1b the references to the left/right panels in the legend seem to be reversed.

We have changed the figure legend in **Extended Data Figure 1b** to

“b, Right: mean GFP fluorescence intensity measured by flow cytometry when transduced HEK cells were treated at 42°C or 43°C for 15, 30 or 45 min. Left: percentage of live cells normalized to the unheated control (i.e., 37°C).”

The concern has been addressed.